# Extraction of wind and temperature information from hybrid 4D-Var assimilation of stratospheric ozone using NAVGEM

Douglas R. Allen, Karl W. Hoppel, David D. Kuhl

Remote Sensing Division, Naval Research Laboratory, Washington, DC, USA

*Correspondence to*: D. R. Allen (douglas.allen@nrl.navy.mil)

**Abstract.**

Extraction of wind and temperature information from stratospheric ozone assimilation is examined within the context of the Navy Global Environmental Model (NAVGEM) hybrid 4D-Var data assimilation (DA) system. Ozone can improve the wind and temperature through two different DA mechanisms: (1) through the "flow-of-the-day" ensemble background error covariance that is blended together with the static background error covariance and (2) via the ozone continuity equation in the tangent linear model and adjoint used for minimizing the cost function. All experiments assimilate actual conventional data in order to maintain a similar realistic troposphere. In the stratosphere, the experiments assimilate simulated ozone and/or radiance observations in various combinations. The simulated observations are constructed for a case study based on a 16-day cycling truth experiment (TE), which is an analysis with no stratospheric observations. The impact of ozone on the analysis is evaluated by comparing the experiments to the TE for the last 6 days, allowing for a 10-day spin-up. Ozone assimilation benefits the wind and temperature when data are of sufficient quality and frequency. For example, assimilation of perfect (no applied error) global hourly ozone data constrains the stratospheric wind and temperature to within ~2 ms$^{-1}$ and ~1 K. This demonstrates that there is dynamical information in the ozone distribution that can potentially be used to improve the stratosphere. This is particularly important for the tropics, where radiance observations have difficulty constraining wind due to breakdown of geostrophic balance. Global ozone assimilation provides the largest benefit when the hybrid blending coefficient is an intermediate value (0.5 was used in this study), rather than 0.0 (no ensemble background error covariance) or 1.0 (no static background error covariance), which is consistent with other hybrid DA studies. When perfect global ozone is assimilated in addition to radiance observations, wind and temperature error decreases of up to ~3 ms$^{-1}$ and ~1 K occur in the tropical upper stratosphere. Assimilation of noisy global ozone (2% errors applied) results in error reductions of ~1 ms$^{-1}$ and ~0.5 K in the tropics and slightly increased temperature errors in the Northern Hemisphere polar region. Reduction of the ozone sampling frequency also reduces the benefit of ozone throughout the stratosphere, with noisy polar-orbiting data having only minor impacts on wind and temperature when assimilated with radiances. An examination of ensemble cross-correlations between ozone and other variables shows that a single ozone observation behaves like a potential vorticity (PV) "charge", or a monopole of PV, with rotation about a vertical axis and vertically oriented temperature dipole. Further understanding of this relationship may help in designing observation systems that would optimize the impact of ozone on the dynamics.

# 1 Introduction

The spatial-temporal variability of long-lived tracers such as stratospheric ozone contains dynamical information that can potentially be exploited to improve analyses of wind and temperature in the stratosphere and mesosphere, where direct wind observations are largely absent. Various studies have examined tracer-wind interactions within a variety of data assimilation (DA) systems including extended Kalman Filter (EKF) (Daley, 1995, 1996), 4D Variational assimilation (4D-Var) (Andersson et al., 1994; Riishøjgaard, 1996; Peuch et al., 2000; Andersson et al., 2007; Peubey and McNally, 2009; Semane et al., 2009; Han and McNally, 2010; Dragani and McNally, 2013; Allen et al., 2013, 2014), Ensemble Kalman Filter (EnKF) (Milewski

and Bourqui, 2011, Allen et al., 2015), and hybrid 4D-Var (Allen et al., 2016). Initial 1D and 2D investigations by Daley (1995, 1996) and Riishøjgaard (1996) and 3D investigations by Peuch et al. (2000), Semane et al. (2009), and Allen et al. (2013) showed that coupling the tracer continuity equation with the dynamical equations could allow wind information to be extracted from tracer observations in either 4D-Var or EKF. These studies illustrated the potential of tracer assimilation to influence winds, but also highlighted limitations on this process from observation quality and sampling, inadequate tracer modeling, and geophysical variability. Further theoretical studies by Allen et al. (2014, 2015, 2016) examined the tracer-wind mechanisms in the shallow water model framework using a hierarchy of DA systems: 4D-Var, EnKF, and hybrid 4D-Var. Additional work by Milewski and Bourqui (2011) examined the assimilation of ozone in the EnKF framework using a 3D data assimilation system, highlighting the propagation of information from ozone to wind via background error covariances.

As seen from these studies, there are two primary ways that ozone can influence winds in hybrid 4D-Var DA. The first way is via the ensemble cross-correlations between ozone and other variables that are blended into the initial background error covariance. These so-called "errors of the day" allow ozone to influence dynamical variables directly. Allen et al. (2016) showed that including these cross-correlations provides additional ozone-wind benefit over conventional 4D-Var that excludes

initial cross-covariances between ozone and other variables. Second, if ozone is included in the cost function, increments to the dynamical fields at the beginning of the analysis window (i.e., strong-constraint 4D-Var), will be adjusted to minimize the differences between the tangent linear ozone forecast and the ozone observations distributed throughout the analysis time window. This linear approximation and adjoint of the tracer continuity equation propagate the ozone sensitivities over the analysis time window (see Allen et al. (2013) for a 1D heuristic analytical solution to this problem to illustrate ozone influence

in 4D-Var). Thereby ozone observations can influence the winds indirectly as the system attempts to reduce the ozone innovations via both wind and ozone increments. Note that two additional ways that ozone assimilation could potentially benefit winds are that improved ozone fields could result in improved radiative calculations in the forecast model (e.g., Cariolle and Morcrette, 2006) as well as improved representation of ozone in forward modeling of ozone-sensitive radiation channels (e.g., Dragani and McNally, 2013); we do not attempt to address these two mechanisms in the current study.

While the potential for tracers to influence winds in DA systems has been well established, the ultimate goal is to obtain operational benefit from this process. Andersson et al. (2007) and Peubey and McNally (2009) showed a tropospheric benefit when infrared and microwave humidity channels from geostationary and polar-orbiting satellites were assimilated in the European Centre for Medium-Range Weather Forecasts (ECMWF) 4D-Var system. As demonstrated by Peubey and McNally

(2009), the dominant factor involves the adjustment of the winds to match observed humidity features (the so-called "tracer advection effect"). Semane et al. (2009) found a slight reduction ($< 0.1$ ms$^{-1}$) in the global wind bias (relative to radiosondes) in the lower stratosphere when assimilating Microwave Limb Sounding (MLS) ozone with the Météo-France 4D-Var system coupled to an offline chemistry transport model. However, other attempts to assimilate stratospheric ozone using 4D-Var algorithms and the resultant dynamical coupling have resulted in problems in operational numerical weather prediction (NWP).

For example, Han and McNally (2010) state that biases between ozone observations and model background led to erroneous wind and temperature increments in the stratospheric analyses in the ECMWF system. These biases could potentially be alleviated by including a bias correction scheme for ozone data (e.g., Dethof and Hólm (2004)), but this (as far as we are aware) has yet to be accomplished. Other potential problems could include breakdown of the tangent linear model in the presence of large ozone gradients (Riishøjgaard, 1996), errors in parameterized ozone photochemistry, and improper characterization of observational and/or forecast model errors. The dynamical benefit of stratospheric ozone assimilation in an operational framework is also challenged by the huge number of additional competing observations from microwave and infrared sounders (on the order of millions of observations per cycle). Whether ozone assimilation can add significant value to operational analyses and forecasts is yet to be determined.

This study attempts to move one step further toward determining whether stratospheric ozone assimilation can benefit analyses in operational NWP models, focusing on ozone-dynamical (i.e., wind and temperature) interactions within a hybrid 4D-Var system, the Navy Global Environmental Model (NAVGEM). Ozone observations have been assimilated in research versions of NAVGEM for some time (e.g., Eckermann et al., 2009). These have produced reliable ozone fields, but the impact of ozone on the dynamics has not been determined, except for a single update cycle demonstration by Allen et al. (2013). An observing system experiment (OSE) could be performed with NAVGEM (as in Semane et al., 2009 with the Météo-France NWP suite and MLS data). We have opted, however, to use simulated ozone observations in this study in order to have controlled experimental conditions. One reason for using simulated observations is to eliminate the impact of biases between observations and model that have caused problems in earlier studies. Another reason is to probe the potential dynamical information content available from stratospheric ozone assimilation by using global sampling patterns with unrealistic temporal and spatial coverage, in addition to the more realistic sampling provided by MLS. The overall goal is to build understanding of how ozone assimilation influences the dynamics that can help in design and interpretation of future OSEs.

Data impact experiments with simulated observations are generally performed using the observing system simulation experiment (OSSE) approach (see Masutani et al., 2010 for a general overview and Timmermans et al., 2015 for a discussion of OSSEs in the context of chemical species). In the traditional OSSE, all data are produced from a model forecast truth (or "Nature Run"). After randomly perturbing the observations in a manner consistent with assumed error statistics, these data are assimilated into a data assimilation system (DAS) in varying combinations. The OSSE approach is used for testing future observations, where data are not yet available. Since the truth is known, the analyses from these experiments can be directly compared to the truth to obtain absolute observation impact. The difficulty is that OSSEs are computationally expensive, requiring the simulation of large numbers of observations of various types. An alternative approach, used by Harnisch et al. (2013), combined real and simulated data in an ensemble of data assimilations (EDA) in order to determine relative impact of future data systems. The real observations included the operational global observing system (GOS), while the simulated Global Navigation Satellite System (GNSS) radio occultation (RO) measurements were simulated from ECMWF analyses. Harnisch

et al. (2013) explain that data simulated in this way are not independent of the real observations, even if a different DAS is used, since there is a common database of observations in the GOS. Therefore, assimilating the real and simulated data together poses some challenges if the observation errors are assumed to be uncorrelated. The analysis of relative changes in ensemble spread is useful for determining the information content of the new observations, but one cannot assess the impact of the new observations on the absolute skill of the EDA mean state with this approach. A similar EDA approach was taken by Tan et al. (2007) to determine the impact of future Atmospheric Dynamics Mission Aeolus wind-profiling LIDAR data.

In this paper, we take a novel approach that combines real and simulated data sets, but in a way that attempts to reduce the error correlations between the two sets. This is performed by spatially separating the data into two regions, nominally defined as troposphere (pressures greater than 100 hPa) and stratosphere (pressures less than 100 hPa). The truth is created from a cycling analysis that assimilates real conventional data in the troposphere, while simulated observations are all located in the stratosphere. This separation is not complete, since vertical background error covariances can extend upward into the stratosphere, and observation weighting functions can extend downward into the troposphere. In addition, separating the troposphere and stratosphere in this way does not perfectly decouple the model fields, since the stratosphere is affected by forcing from below and vice-versa. However, this dynamical coupling is expected to weaken with increased altitude. This is illustrated in the paper by widely divergent stratospheres resulting from simulations using the same tropospheric observations, but using unperturbed and perturbed stratospheric initial conditions. As far as the decoupling assumptions being valid, this approach allows us to calculate absolute impact, since the truth is known, but avoids having to perform a complete OSSE.

Simulated stratospheric observations (ozone profiles and microwave and infrared radiance measurements) are created using the truth. In addition to simulating ozone observations from a typical polar-orbiting limb sounder, we also examine the impact of globally distributed ozone observations using two different sampling patterns in order to explore the information content available in the stratospheric ozone fields. The ozone data are assimilated with and without simulated satellite radiance measurements in order to determine the impact on the stratosphere with and without a realistic GOS. The impact of ozone on the wind and temperature analyses is examined for a case study in the Northern Hemisphere late fall, 15 November - 1 December 2014. Examining this case study allows us to lay the framework for future work to assess the statistical impact of ozone assimilation over longer periods.

The paper is outlined as follows. The NAVGEM NWP system is described in Sect. 2. The characteristics of the ensemble cross-covariances are examined in Sect. 3 in order to understand how ozone and other variables relate. Section 4 explores how well ozone by itself can constrain the stratospheric dynamics. An examination of ozone assimilation in the presence of simulated radiance observations is presented in Sect. 5, and Sect. 6 provides a summary and conclusions.

## 2. Model description

### 2.1 Forecast model

This study uses a reduced resolution version of the operational NAVGEM described in Hogan et al. (2014). The NAVGEM global forecast model uses a semi-Lagrangian/semi-implicit integration of the hydrostatic equation, the first law of thermodynamics, and conservation of moisture and ozone. This study uses a 60 level hybrid sigma-pressure coordinate (top at 0.05 hPa) as described in Eckermann et al. (2009). There are 18 levels in the stratosphere and 7 in the lower mesosphere (defined in this study as pressures ranging from 1.0 to 0.05 hPa), and the vertical spacing ranges from ~1.5 km in the lower stratosphere to ~2.5 km at the stratopause to ~5 km at the model top. The model is run at a relatively low resolution of T47 (144 longitudes × 72 latitudes, for a Gaussian grid spacing of ~2.5° at the Equator). The model time step is 1800 s. The same forecast configuration and resolution are used for the control (outer loop) and the ensemble forecasts and DA (inner loop). The current (early 2018) operational resolution uses the same 60 vertical levels and a horizontal resolution of T425 (Gaussian grid spacing of ~0.28° at the Equator) for the outer loop and T119 (Gaussian grid spacing of ~1.0° at the Equator) for the innner loop.

### 2.2 Hybrid 4D-Var data assimilation system

The NAVGEM DA system employs a hybrid 4D-Var method, which is becoming increasingly popular at operational NWP centers (e.g., Buehner et al., 2010; Bonavita et al., 2012; Clayton et al., 2013; Kuhl et al., 2013; Kleist and Ide, 2015). NAVGEM minimizes a quadratic cost function using the accelerated representer approach as described in Xu et al. (2005) and Rosmond and Xu (2006). The conventional initial background error covariance $\mathbf{B}_0^{con}$ is calculated using an analytic formulation that employs the hydrostatic relationship in the vertical between geopotential and temperature, and wind-geopotential correlations based on approximate geostrophic balance on an *f*-plane, i.e., constant Coriolis parameter with latitude (Daley, 1991; Daley and Barker, 2001; Kuhl et al., 2013). There is no coupling between ozone and dynamical variables in $\mathbf{B}_0^{con}$, but coupling between these variables does develop implicitly over the 4D-Var time window. The only difference between the hybrid 4D-Var used in this study and Kuhl et al. (2013) is the incorporation of ozone observations in the analysis and ozone in the ensemble forecasts and forecast error covariance.

The tangent linear model (TLM) currently used in NAVGEM is based on linearization of the Navy Operational Global Atmospheric Prediction System (NOGAPS) global spectral forecast model (Hogan et al., 1991), which was the forerunner of NAVGEM. Relevant details of the TLM and adjoint (ADJ) used in this study are provided in Rosmond (1997). The TLM and ADJ are also run at T47 resolution with 60 vertical levels, as is the nonlinear forecast model, but with a reduced time step of 900 s. The TLM has parameterizations for surface flux and vertical mixing based on Louis (1979), but does not include other physical parameterizations such as radiation, ozone chemistry, and gravity wave drag, which are in the nonlinear model. The

4D-Var system runs with a 6-h analysis window, and the analysis at the middle of one window is used to initialize a 9-h forecast that serves as the background for the next update cycle. Each analysis and resulting 1-5 h forecast are saved for use in creating the simulated observations.

The ensemble consists of 80 members, which are updated each cycle using the ensemble transform (ET) approach described by McLay et al. (2008, 2010) and Kuhl et al. (2013). The ET scheme transforms the previous 6-h ensemble perturbations into a new set of initial perturbations such that the initial ensemble covariance is consistent with a prescribed climatological 3D-Var based estimate of the analysis error variance; this is not a Kalman filter DA scheme and does not require observations. The climatological variances are averaged from 10 June 2015 to 10 August 2015, and are the same as those used in the operational

NAVGEM system. The ensemble covariance, $\mathbf{B}_0^{\text{ens}} = \mathbf{X}'\mathbf{X}'^{\text{T}} / (N_{\text{ens}} - 1)$, is calculated at the start of each 6-hour window using the ensemble states $\mathbf{X}$; the prime indicates perturbation from the ensemble mean, the superscript T indicates transpose, and $N_{\text{ens}}$ is the ensemble size. The ensemble covariance is then blended together with $\mathbf{B}_0^{\text{con}}$ using $\mathbf{B}_0^{\text{hybrid}} = (1 - \alpha)\mathbf{B}_0^{\text{con}} + \alpha \mathbf{S} \circ \mathbf{B}_0^{\text{ens}}$, where $\alpha$ is a blending coefficient between zero and one, $\mathbf{S}$ is the localization function, and the open circle indicates the Schur product. The horizontal localization is based on a second-order autoregressive

function and the vertical localization employs a Gaussian log-sigma correlation as described in Daley and Barker (2001) and Kuhl et al. (2013). The ensemble was initialized using NAVGEM analyses from a separate experiment that included conventional data along with radiances and MLS ozone and temperature profiles. This separate experiment was run at T119 with 74 levels and was downscaled to the T47L60 resolution used for this study. The data were sampled at 18 Z on 80 consecutive days, starting 2 May 2014. Ensemble standard deviations are examined in Sect. 3.1.

**2.3 Observations**

The experiments in this study assimilate both actual observations (in the troposphere) and simulated stratospheric observations computed from the truth experiment (described in Sect. 2.4). The tropospheric observations include conventional operational measurements, such as surface observations from ships, buoys, and land surface stations, upper air observations from radiosondes and aircraft, and satellite-derived winds. Global Positioning System (GPS) radio occultation observations are not

included for this particular study. The standard NAVGEM operational data quality control and thinning algorithms are used, and the resulting tropospheric observation counts range from ~750,000 to 1,000,000 observations for each 6-h cycle. The actual observations are limited to pressures greater than 100 hPa, which mainly affects the radiosonde profiles.

For the simulated ozone observations, three different sampling patterns are used: global, polar-orbiting, and random (See Fig. 1). The global observations (Fig. 1a) are provided on an approximately equal-area sampling, generated by subdividing an

icosahedral base into a triangular grid with ~300 km spacing (3840 elements). To avoid both horizontal and vertical

interpolation by the DA, the ozone observation locations were moved to the nearest model Gaussian grid points, and the ozone was sampled vertically on the model levels. Seventeen vertical levels in the stratosphere are used, ranging from 77 to 1.1 hPa (~20 to 50 km altitude). Note that for the 60-level NAVGEM configuration, these stratospheric levels are all constant pressure levels. The temporal sampling for the global observations is 1 h, matching the forecast output sampling. The second set of ozone observations simulates a polar-orbiting limb sounder, and was created by sampling the TE at the observation locations of the Aura MLS instrument. There are approximately 3500 observations per day for MLS; a sample flight track is shown in Fig. 1b. For the polar-orbiting data, we also move the locations to the nearest model grid point and sample vertically on the same model levels. The third set of ozone observations (Fig. 1c), which we call "random," sub-samples the global observations randomly in space and time at a frequency of 3500 per day, which is similar to that of the polar-orbiting observations. This tests whether spreading the information from polar-orbiting sampling would make the data more useful for extracting wind and temperature information.

For the assimilation of these ozone observations, the observation error covariance is uncorrelated with a specified standard deviation $\sigma_{ob}$. We will examine cases with "perfect" (i.e., no random error added) and "noisy" (with applied random error) observations. Note that for the perfect observations, we do not set $\sigma_{ob}$ exactly to zero, but to a reasonably small value of 0.1 ppmv. Setting $\sigma_{ob}$ to exactly zero causes the cost function to become singular and prevents the solution from converging. Calling these observations "perfect," while having non-zero observation error in the DAS, follows the naming convention of Peuch et al. (2000). For the "noisy" observations, we apply 2%, 5%, or 10% random error to the perfect observations, and we set $\sigma_{ob}$ to the same percent value, except that we limit $\sigma_{ob}$ to a minimum value 0.1 ppmv. The lower limit is to prevent the DAS from constraining too tightly to highly precise ozone observations, which could result in spurious wind increments.

Some of the experiments assimilate simulated stratospheric radiance observations, in order to assess the value of ozone in combination with a typical set of global observations. NAVGEM routinely assimilates microwave and infrared radiances from a number of sounders. In this study, simulated radiances are created using actual data sampling from Advanced-Microwave Sounding Unit (AMSU-A), Atmospheric Infrared Sounder (AIRS), Advanced Technology Microwave Limb Sounder (ATMS), and Infrared Atmospheric Sounding Interferometer (IASI) (see Fig. 2 for an example of radiance observations for one update cycle). During the creation of the truth experiment, actual radiance observations are processed to the point where background radiance values are calculated, but are then omitted from the DA solver. The background radiance values then become "perfect" simulated radiances. Because the simulated radiances are created and then assimilated using same radiative transfer model, they are unbiased "perfect model" data. For the radiance and ozone assimilation experiments (Sect. 5), we add Gaussian random noise to the perfect radiance data, matching the observation error values used in NAVGEM for the actual instruments. These provides realistic "noisy" radiance observations. The variational radiance bias correction scheme in the DA

is disabled for the simulated radiances. This is a best-case scenario for the radiances, since the DA of true radiances always includes biases. Addressing the impact of bias correction in the context of ozone assimilation is beyond the scope of this paper.

As explained in the Sect. 1, the separation of the troposphere and stratosphere is not perfect, since tropospheric observations, as well as specified forecast errors, can have vertical error correlations that extend upward into the stratosphere. For example, tropospheric temperature increments can raise the geopotential height at altitudes in the stratosphere, and wind/height balances in the conventional and ensemble covariances will result in stratospheric circulation increments. These, in turn, can influence the stratospheric ozone field via advection. In addition, the adjoint and tangent linear integrations can propagate information vertically to and from the observation locations. To illustrate the combined effects of these non-local processes, Fig. 3a,b shows the time mean and zonal mean of the absolute value of the $u$ and $T$ increments over the 6-day period, 25 November - 1 December 2014 from the truth experiment (described in the next section). While all observation point locations for the truth experiment are located below 100 hPa, small increments extend above 100 hPa. The increments in the stratosphere are limited to less than around 1.0 ms$^{-1}$ for zonal wind and ~0.4 K for temperature. The stratospheric increments to $u$ and $T$ are larger in the NH than in the SH, likely due to more radiosondes over the continental regions. The largest wind increments are in the tropical upper troposphere, where we also see the largest ozone increments (plotted in Fig. 3c as percentage of the local analyzed ozone). The ozone increments are small in the region of simulated ozone observations (region between the white dashed lines on Fig. 3), generally less than 1%, suggesting that the coupling will not adversely affect our ozone assimilation experiments. The simulated radiances will have a weak correlation with the tropospheric observations from the temperature increments in the stratosphere, as well as from radiance weighting functions that extend into the troposphere. The wind increments will not directly influence the simulated radiance and ozone observations, but may affect them indirectly via advection over the assimilation window.

## 2.4 Truth experiment and meteorological conditions

The experimental design (described in more detail in Sect. 2.5) is based on a truth experiment (TE) that is used to simulate observations that are assimilated back into the system and to evaluate all other experiments. This TE could be created by a free-running nature forecast, which would be used to simulate a full set of conventional tropospheric observations, as in conventional OSSEs. As explained in Sect. 1, we took a simpler approach in which our TE is a normal cycling analysis in which only tropospheric data (pressures greater than 100 hPa) are assimilated. All subsequent experiments, with differing stratospheric observations, assimilate the same set of tropospheric observations. This gives similar (but not exactly the same, as discussed below) tropospheric analyses for each experiment, but the stratospheric analyses can vary widely. This approach provides a realistic evaluation of how differing stratospheric observations impact a typical global analysis. We note here (and will show later) that when stratospheric observations are included, the analyzed tropospheric state will be different from the TE. As explained by Geer (2016), even a slight numerical perturbation will generate chaotic divergence between two analyses. In Sect. 4.1 we will examine the extent to which the troposphere is sensitive to "perturbations" in the data set being assimilated.

The stratospheric truth should therefore be considered as one realization drawn from a potential ensemble of stratospheric states, given the potential variation in the troposphere caused by these slight perturbations.

We initialized the TE on 15 November 2014 (00Z) and ran through 1 December 2014 (00Z), for a total of 16 days with 64 cycles. The initial zonal mean latitude/pressure cross-sections for zonal wind, temperature, and ozone are plotted on the top row of Fig. 4. The Arctic winter stratospheric vortex is seen in the Northern Hemisphere (NH), extending to the top of the model with peak winds of ~70 ms$^{-1}$. In the polar region of the Southern Hemisphere (SH), westerlies occur in the lower stratosphere and easterlies occur in the upper stratosphere and lower mesosphere. The westerlies are the remnant of the Antarctic winter stratospheric vortex, which is in the process of breaking down. In the tropics, a complicated pattern of alternating easterlies and westerlies is observed, with a large region of easterlies in the lower stratosphere. The zonal mean temperature shows typical solstice conditions, with warm summer and cold winter stratosphere, a cold tropical tropopause, and a warm troposphere. The ozone mixing ratio maximizes in the tropical middle stratosphere, with the peak shifted towards the SH. Low ozone mixing ratios occur in the troposphere, mesosphere, and SH polar lower stratosphere (i.e., ozone hole conditions).

The meteorological conditions for the lower, middle, and upper stratosphere are illustrated in Figs. 5, 6, and 7, which show the ozone and geopotential height at 77 hPa (~18 km), 10.5 hPa (~32 km), and 1.1 hPa (~48 km), respectively, from 15-30 Nov 2014 (all at 00Z) at 3-day intervals. At 77 hPa (Fig. 5), the NH stratospheric polar vortex can be identified by closed height contours, with the center displaced off the pole towards Asia. The NH ozone has generally higher mixing ratio in the vortex and lower in the tropics at this level. Over the course of the next two weeks, the vortex shape is modulated by ridges forming on both sides of the vortex, resulting in a dumbbell-shape on 30 November. This dynamical activity is accompanied by ozone advection eastward and northward from the tropics (for example, see the tongue of low ozone air forming at the bottom of Fig. 5e). High ozone occurs along the edge of the vortex, and the mixing ratio increases with time over late November. In the SH, ozone depletion is evident within the Antarctic vortex, while higher ozone occurs in the extratropics. The Antarctic vortex shifts over the course of late November, being drawn into an oval shape by the end of the month and displaced well off the pole. The low ozone contours follow the shape of the vortex over this period.

At 10.5 hPa (Fig. 6) the NH polar vortex, indicated by low ozone mixing ratio and enclosed height contours, is seen on 15 November centered slightly off the pole. A tongue of low-latitude air moves northward and eastward over the next few days as an Aleutian high starts to spin up, pushing the vortex off the pole. The vortex elongates and the minimum ozone mixing ratio increases, indicating some mixing of the vortex air outward. The Aleutian high is still strong on 30 November. In the SH, the large-scale circulation in the middle stratosphere is generally becoming easterly in November, with accompanying high pressure and lower ozone mixing ratio. On 15 November, a low pressure cyclone between South America and Antarctica disrupts the otherwise easterly flow. Low latitude air with high ozone is pulled clockwise around the cyclone on 18 and 21

November. The cyclone diminishes in strength from 21-30 November, and the prevailing anticyclonic flow center moves back towards the pole.

At 1.1 hPa (Fig. 7), The Arctic vortex is initially centered close to the pole, with low ozone inside. The ozone mixing ratio in the vortex increases sharply by 18 November. This is likely due to parameterized photochemistry drawing the ozone towards a climatological state that was different from the simulation used for the initial conditions. The upward extension of the growing Aleutian high is seen at this level as well, forcing the vortex off the pole and stretching it into a "comma" shape by 27 November. In the SH, the height contours are nearly zonal throughout this period, with steady easterly circulation, and the ozone becomes nearly zonal as well. This suggests that the wave activity observed at 10.5 hPa becomes trapped before it reaches the upper stratosphere, due to the presence of the zero wind line (see Fig. 4a), which serves as a critical line for stationary planetary waves. The overall meteorological situation for this period is characterized by a decaying Antarctic vortex in the lower stratosphere and quiescent SH easterlies in the upper stratosphere/lower mesosphere, while in the NH, the Arctic vortex is being influenced by moderate wave activity that is causing the vortex to be pushed off the pole and stretched. This provides a range of dynamical conditions for testing the impact of ozone assimilation on the winds and temperature.

## 2.5 Experimental design

We perform two types of stratospheric assimilation experiments: ozone-only assimilation and ozone/radiance assimilation. To illustrate the differences between these experiments, Fig. 8 provides schematic diagrams. Both types of experiment use the same TE and observation database (note that all experiments assimilate the same tropospheric data). Except for the TE and a few test experiments, most of the experiments use unperturbed initial conditions in the troposphere and perturbed initial conditions in the stratosphere. The perturbation is performed by replacing the initial conditions with a different stratospheric analysis (at pressures less than 100 hPa), valid at the same time (15 November 2014, 00Z), but based on NAVGEM experiments that differ in terms of model resolution and data assimilated, resulting in slightly different dynamical fields. The ozone fields, however, are initially identical. Figure 4 (second row) shows the zonal mean cross-sections of zonal wind, temperature, and ozone for the perturbed state, and Fig. 4 (third row) shows the differences between perturbed and unperturbed initial conditions. Large dynamical differences occur in the tropical upper stratosphere and throughout the lower mesosphere.

For the ozone-only assimilation (Fig. 8, top row, results presented in Sect. 4), a baseline experiment (BE) is performed by running the system from the perturbed initial conditions and assimilating only the tropospheric conventional observations. As will be shown below, the stratospheric winds and temperature in this BE deviate significantly from the TE (after 16 days, zonal mean differences of up to ~80 ms$^{-1}$ occur for vector wind and ~25 K for temperature). When experimenting with the blending coefficient ($\alpha$ = 0.0, 0.5, and 1.0), we must run separate TE and BE cases for each value. This is because the blending coefficient affects the tropospheric assimilation, and hence changes the reference TE for each case. Changing $\alpha$ examines the sensitivity of the amount of ozone-wind correlation being used from the ensemble covariances. Ozone-only assimilation

experiments are next performed, which examine the limit to which ozone could potentially constrain the winds without any other data present in the stratosphere. Global data are assimilated for all three values of α (Sect. 4.2), while random and polar-orbiting data are only assimilated for α = 0.5 (Sect. 4.3). In addition, to examine sensitivity to data quality, experiments are performed for α = 0.5 with assimilation of ozone data with imposed observational errors (Sect. 4.4).

In the ozone/radiance assimilation experiments (Fig. 8, bottom row, results presented in Sect. 5), we test the extent to which assimilating ozone data can reduce the errors relative to a system constrained by realistic radiance observations. First, we create a BE by assimilating noisy radiances created from the TE. Then, experiments are performed in which either global, random, or polar-orbiting ozone data are assimilated in addition to the radiances using either perfect ozone or ozone with

imposed random errors. These experiments are all run for 16 days, which allows for a 10-d spin-up of the ensemble and errors (discussed below) and evaluation of experimental errors over 6 days. This single case study does not include enough data to adequately assess statistical significance (e.g., Geer, 2016), but will provide a framework for analyzing and guidance in designing future long-range experiments.

## 3. Discussion of background errors

The background error covariance is a critical component of the hybrid 4D-Var system. Hybrid 4D-Var combines a conventional error covariance with a localized ensemble covariance in order to take advantage of both the high-rank properties of the conventional and flow-of-the-day properties of the ensemble components. In this section, we first examine the latitude/pressure cross-sections of the conventional and ensemble errors (Sect. 3.1) and the ensemble spin-up (Sect. 3.2). We next examine horizontal maps of the ensemble background error standard deviations (i.e., the square root of the diagonal terms of the

covariance matrix) in Sect. 3.3. Finally, in Section 3.4, we examine the cross-correlation terms, which indicate how errors are correlated with other variables and spatial locations. Our particular interest is the patterns that describe how ozone correlates with other variables. For simplification, we denote the background error standard deviations as $\sigma_{con}$, $\sigma_{ens}$, and $\sigma_{hyb}$ for the conventional, ensemble, and hybrid, respectively.

## 3.1 Comparing conventional and ensemble error standard deviation

Figure 9 shows latitude/pressure cross-sections of $\sigma_{con}$ and $\sigma_{ens}$ for zonal wind ($u$), temperature ($T$), and ozone ($O_3$). The $\sigma_{ens}$ has been zonally averaged, while $\sigma_{con}$ is formulated as a zonal mean model. The conventional errors are shown for 15 November 2014 (00Z) only, while the ensemble errors are provided for 15 November and 1 December 2014 (both at 00Z). The $\sigma_{con}$ for zonal wind (Fig. 9a) increases with altitude from ~2 ms$^{-1}$ in the troposphere to ~8 ms$^{-1}$ at 0.1 hPa. The $\sigma_{ens}$ for zonal wind on 15 November (Fig. 9d) shows more structure, with higher values in the upper troposphere and lower mesosphere,

and lower values in the extratropical stratosphere. As the ensemble evolves over the next 16 days, the zonal wind $\sigma_{ens}$ generally increases, particularly in the lower mesosphere, as seen in Fig. 9g, peaking at over 15 ms$^{-1}$.

The temperature $\sigma_{con}$ (Fig. 9b) ranges from ~0.5 to ~2 K, with lower values in the tropics. The initial $\sigma_{ens}$ for temperature (Fig. 9e) has a similar geographic structure as zonal wind, with elevated values in the upper troposphere and mesosphere. The temperature $\sigma_{ens}$ generally increases with time, with large values (>5 K) occurring above ~1 hPa on 1 December (Fig. 9h). For ozone, the $\sigma_{con}$ is prescribed as a constant value of 0.3 ppmv, except for elevated values in the tropical troposphere (Fig. 9c). The initial ozone $\sigma_{ens}$ (Fig. 9f) is elevated in the tropical middle stratosphere and SH lower and upper stratosphere. The ozone errors evolve by 1 December to have three regions of enhanced $\sigma_{ens}$ located in the middle stratosphere in the tropics and extratropics of each hemisphere, with relative minima in the subtropics (Fig. 9i). Lower values are seen in both the troposphere and upper stratosphere/mesosphere, which largely reflect the lower ozone mixing ratios in these levels.

## 3.2 Ensemble spin-up

To determine when the ensemble has finished its spin-up phase, we calculated the globally averaged $\sigma_{ens}$ for $u$, $T$, geopotential height ($Z$), and ozone at each model level and then vertically averaged over all levels. For the vertical average, we weighted the profile by a layer thickness in km. The thickness was calculated by first choosing a nominal pressure value for each NAVGEM level, based on a model pressure profile with surface value of 1000 hPa, then calculating the log-pressure height for each level using a constant scale height of 7 km, and finally differencing adjacent layers to get thicknesses. The resulting time-series plots (daily values at 00Z from 15 November - 1 December 2014) in Figure 10 show that the first decrease in spread for $u$ (similar results were obtained for meridional wind, $v$) occurs on 25 November (after 10 days of cycling), while for $T$ and $Z$ the first decrease occurs on day 23 November (8 days). We therefore consider the ensemble to be spun up by 25 November. The ozone $\sigma_{ens}$ increases monotonically from 16 November - 1 December, and therefore we cannot assign an objective spin-up time using this approach. However the rate of increase becomes quite small by 25 November (less than 1.5% per day), so we consider the ensemble dependence on the initial ozone ensemble to be small after 10 days of cycling. Further examination of the evolution of ozone $\sigma_{ens}$ will be presented in Sect. 3.3. Neglecting the 10-day spin-up, error results presented in Sections 4 and 5 are generally averages over the last 6 days (25 November, 00Z - 1 December 2014, 00Z), which includes 25 separate analyses.

### 3.3 Horizontal maps of ozone $\sigma_{ens}$

The ozone $\sigma_{ens}$ shows strong geographic patterns that are related to the flow-of-the-day. Figure 11 shows horizontal maps of the ozone $\sigma_{ens}$ for the same level (10.5 hPa) and dates as in Fig. 6. Geopotential contours are overlaid on the $\sigma_{ens}$ to facilitate comparison with the flow. On 15 November, the initial $\sigma_{ens}$ is not aligned with the flow, since the initial ensemble was constructed with analyses on consecutive days from an offline experiment; after 3 days, however, flow-like patterns start to emerge. On 18 November, the $\sigma_{ens}$ in the NH is larger within the polar vortex, while smaller values occur outside of the vortex. This pattern strengthens over the next several days so that by 24 November the vortex/extra vortex distinction is prominent. The high $\sigma_{ens}$ in the vortex is in a location where the ozone mixing ratio is actually low (see Fig. 6). While individual maps of ensemble members (not shown) indicate some variability in the location, orientation, and shape of the vortex, the ozone maps exhibit even larger variability. We think this is due to slight variations in the vortex evolution over time in each ensemble member that result in differences in ozone advection that accumulate with time due to the long photochemical lifetime of ozone in the NH winter polar region. This process causes the initially small ozone spread to increase over the experiment (as seen in Fig. 10d). Further work is necessary to elucidate the exact mechanisms that force changes in ozone ensemble spread. Long streamers of high $\sigma_{ens}$ are visible in the NH throughout this period, circling around the outer edges of the polar vortex and Aleutian high, where the ozone gradients are large. These patterns are significant for data assimilation, since they will affect the weight that is given to the observations. For example, in the polar vortex the $\sigma_{ens}$ is large, so ozone observations in this region would be expected to have a larger impact than those in regions of low $\sigma_{ens}$.

The ozone $\sigma_{ens}$ in the SH also shows a rapid spin-up from an initial state that is approximately constant in the zonal direction. Flow-dependent patterns are seen on 18 November in the cyclonic region between South America and Antarctica, with a low $\sigma_{ens}$ "tongue" surrounded by high $\sigma_{ens}$. From 18 to 27 November, the $\sigma_{ens}$ in the anticyclonic closed height contours increases. A tongue of low $\sigma_{ens}$ occurs between the two flow regimes on 24 and 27 November, apparently advected by the nearly cross-polar flow. By 30 November, the $\sigma_{ens}$ pattern shows generally large values at high latitudes, small values in the tropics, and complicated structure in the extratropics. While a complete analysis of the causes of these features is beyond the scope of this paper, it is clear that the $\sigma_{ens}$ is strongly flow-dependent and may (at least in the experiments with $\alpha = 1.0$) result in large differences in weighting of ozone observations. Note that the errors have a large dynamic range from ~0.04 to 0.99 ppmv at 10.5 hPa for 30 November.

### 3.4 Ensemble ozone-wind cross-covariances

As discussed in Sect. 1, ozone can influence wind and temperature via the ensemble background error cross-correlations. Here we show an example of these cross-correlations. Figure 12 provides a composite view of the impact of a single ozone observation at pressure of 10.5 hPa and latitude of 28.6° S. The composite was created by separately calculating the spatial correlations of ozone with all other points and variables at 36 longitudes (0°, 10°, 20°, …, 350°). The correlations were then shifted to a common longitude of 180° E and averaged to reduce spurious noise. The top row of Fig. 12 shows the horizontal correlations. The ozone-ozone correlation (Fig. 12a) has a maximum of 1.0 at the observation point, and then decreases gradually in each direction, with a larger decorrelation length in the zonal direction. The ozone correlates strongly with vorticity (Fig. 12b), with the ozone-vorticity correlation having a similar zonally oriented shape. The ozone-height correlation (Fig. 12c) is more isotropic and represents an anticyclonic circulation, which is counter clockwise in the SH, as seen in the correlations with zonal (Fig. 12d) and meridional (Fig. 12e) wind. The ozone-temperature correlation (Fig. 12f) is weak at the level of the observation, but vertical cross-sections in longitude (Fig. 12l) and latitude (Fig. 12r) reveal a strong dipole pattern with cold (warm) temperature above (below) the observation. Vorticity (Fig. 12h, n) and height (Fig. 12l, o) correlations are vertically oriented similar to the ozone-ozone correlation, with slight westward and southward tilting with height; the wind cross-sections (Fig. 12j, k, p, q) show that the anticyclonic circulation extends above and below the observation.

The temperature and circulation patterns revealed in the correlations of Fig. 12 are similar to those associated with the potential vorticity (PV) "charge" concept developed by Bishop and Thorpe (1994). In this analogy to electrostatics, an elementary PV charge is associated with a field that produces a circulation about the vertical axis and a vertically oriented temperature dipole (see also Fig. 14 of Allen et al., 1997). That a single ozone observation would produce the same circulation patterns as a monopole of PV makes sense, since PV and ozone are both quasi-conserved quantities and will therefore have strong correlations. The pattern is also seen at other latitudes, although its strength varies due to differing ozone gradients and geostrophic coupling. In the NH, the ozone-vorticity correlation is negative and the circulation is clockwise (anticyclonic in the NH), and the temperature dipole is similar with cold (warm) temperature above (below) the observations. These results indicate that ozone observations may be considered as pseudo-PV observations, at least in the regions of strong horizontal ozone gradients. As seen in the zonal mean ozone (Fig. 4c), the ozone contours are approximately vertically oriented at this latitude (indicated by white dashed line), so the horizontal ozone gradients are relatively strong.

### 4. Ozone-only assimilation

In this section, we evaluate the influence of ozone-only assimilation on the wind and temperature analyses. There are several factors that will affect the ozone-wind/temperature relationships in the system. The experiments focus on the sensitivity to perturbations in the DAS, blending coefficient, sampling pattern, and observation error (see Table 1 for a complete list of experiments). To quantify the ozone impact, we calculate the root mean square (RMS) error profiles for the background and

Ozone Assimilation Experiments (OAE) for vector wind and $T$ in three latitude bands (NH, 90°N - 30°N; TR, 30°N - 30°S; SH, 30°S - 90°S). These were calculated by first computing the RMS error (RMSE) for $u$, $v$, and $T$ using the following formula (shown below for $u$, but similar for $v$ and $T$).

$$u_{RMSE}^2(k) = \frac{\sum_{j=j\min}^{j\max}\left[\frac{1}{nlon}\sum_{i=1}^{nlon}\left(u(i,j,k)-u_{TE}(i,j,k)\right)^2 \cos(lat(j))\right]}{\sum_{j=j\min}^{j\max}\cos(lat(j))} \tag{1}$$

Here $i$, $j$, and $k$ are indices for longitude ($lon$), latitude ($lat$), and vertical level, while $nlon$ indicates the number of longitudes and $j$min and $j$max indicate the latitude indices corresponding to the bounding latitudes for each region (NH, TR, or SH). To calculate the vector wind error, we combine the $u$ and $v$ errors as follows.

$$V_{RMSE}^2(k) = \sqrt{u_{RMSE}^2(k) + v_{RMSE}^2(k)} \tag{2}$$

To reduce random noise, the errors are averaged over the last 6 days of the experiment (25 November 2014, 00Z - 1 December 2014, 00Z), allowing for the ensemble spin-up as well as reduction of initial errors. Time series of the vertically averaged vector wind and temperature errors (using the same approach as in Sect. 3.2, but limited to the pressure range of the ozone observations, 77 - 1.1 hPa) are provided in Fig. 13. These show results for the assimilation of ozone and radiances in different combinations, which will be examined in more detail below. Here we simply point out that the errors in each experiment level out around 25 November. We can therefore consider that the sensitivity to initial conditions has been generally lost by 25 November, and the 6-day average is reasonable. Due to limited independent analyses in this study, we do not actually test for statistical significance; Geer et al. (2016) suggest that large numbers (on the order of several hundred) of independent tests may be necessary to determine statistical significance to changes in a NWP system.

**4.1 Sensitivity of the analysis to perturbations in the DAS**

Before comparing the results from the various ozone assimilation experiments, we first examine the dependence of the analysis on perturbations to the system. As discussed by Geer (2016), a perturbation to the observational dataset or the numerics of the system will generate chaotic divergence in a given DAS. By running the perturbed and unperturbed system, we can estimate the analysis variance, which is referred to as the "null hypothesis" by Geer (2016). We generate a simple four-member null set as the TE and experiments that assimilate perfect global, random, or polar-orbiting stratospheric ozone observations, starting from unperturbed initial conditions (see Table 1, Sect. 4.1, for further details on these experiments). For an ideal DA system starting from perfect initial conditions, adding perfect observations would produce zero innovations, which would have no effect on the analysis. However, the inclusion of additional observations relative to the TE changes the numerics of the cost

function minimization, resulting in slightly different analysis increments. Also, the specified ozone $\sigma_{ob}$ standard deviation is not zero (0.1 ppmv), allowing some variation in the realized state. In addition, sub-optimalities in the data assimilation system, such as sampling error in the ensemble background error covariances, may also result in changes to the analyses. These behaviors of the system create a limit on the level of errors we can reliably distinguish from the TE. We also note that small changes in the troposphere will lead to differences in gravity waves that grow exponentially with increasing altitude (see also Sect. 4.3). For a high-altitude analysis, such as the one used here, this should cause the chaotic divergence, or variance, to be largest in the lower mesosphere (Liu et al, 2008).

The results are illustrated in Fig. 14, which shows error profiles averaged over the last 6 days of the experiments. The vector wind and $T$ errors are very similar below about 300 hPa, suggesting that the lower tropospheric response is independent of the choice of stratospheric perturbations. These tropospheric errors could be considered a rough estimate of the spread in a hypothetical tropospheric ensemble created by a large number of analyses with slightly different perturbations. As expected, the differences relative to the TE increase with altitude in the stratosphere and lower mesosphere. In the stratosphere, there are larger variations in errors among the three experiments, with wind errors ranging from ~0.5 - 2 ms$^{-1}$ and $T$ errors ranging from ~0.1 - 1 K. The global experiment gives the largest differences from the TE in the lower stratosphere. This might be expected, since it has the largest number of observations. However, the global experiment has generally smaller errors in the lower mesosphere. This may due to the dense spatial and temporal coverage allowing the best restraint of the error growth from gravity waves. The sparse coverage from polar ozone assimilation, on the other hand, shows the greatest differences from the TE at high altitudes, since it likely has the least ability to limit forecast divergence. Further analysis (beyond the scope of this paper) would be necessary to evaluate this mechanism in detail. Due to the limited number of independent analyses in this study, we do not use this null set to determine statistical significance. However, these results do provide a preliminary estimate of the sensitivity of the analyses to perturbations in the DA system, and the maximum stratospheric errors in Fig. 14 could be considered a rough estimate of the minimum possible errors for our subsequent ozone assimilation experiments.

## 4.2 Dependence on blending coefficient

The next set of experiments assimilate perfect global ozone data in the stratosphere, using blending coefficients of $\alpha = 0.0, 0.5$, and 1.0. For each choice of blending coefficient, three experiments are completed for the TE, BE (no ozone), and OAE (ozone); see nine experiments listed for this subsection in Table 1. As discussed above, separate TE and BE are necessary for each case, since the blending coefficient affects not only the stratosphere, but also the tropospheric analysis. Figure 15 shows RMSE for the BE (dotted) and OAE (solid) for the three blending coefficients. The BE wind errors increase with altitude throughout the stratosphere, ranging from ~1-2 ms$^{-1}$ at 100 hPa to ~60-70 ms$^{-1}$ at 1 hPa in the NH, ~20-25 ms$^{-1}$ in the tropics, and ~3-5 ms$^{-1}$ in the SH. The differences in wind errors in different latitude bands reflect different sensitivities to perturbations in the initial conditions. Due to the low sensitivity in the SH, our discussion will focus mainly on the NH and TR. The BE errors for different

blending coefficients show slight differences, indicating the sensitivity of the stratosphere to changes in the DA system used for tropospheric analysis. The BE $T$ errors are also largest in the NH and TR, with generally increasing errors with height in the stratosphere. BE $T$ errors in the stratosphere reach up to ~20-25 K in the NH, ~5-6 K in the TR and ~1.5-2 K in the SH.

For the OAEs, there are generally large reductions in vector wind errors throughout the stratosphere and mesosphere relative to the BE. In the NH and TR, the results with non-zero blending coefficients are better than with $\alpha = 0.0$ above about 10 hPa. This indicates that the ensemble correlations are playing a large role at higher altitudes. This is expected, since the conventional balance approximations were designed to simulate tropospheric balance conditions, and they do not take into account the influence of resolved unbalanced modes such as gravity waves. The $\alpha = 0.5$ results are slightly better than the $\alpha = 1.0$ results, suggesting that combining covariances is helpful for the system, a well-documented result in the stratosphere (e.g., Kuhl et al., 2013). The $T$ errors for the OAEs also show reductions in the NH relative to the BE, with $\alpha = 0.5$ producing the consistently smallest errors throughout the stratosphere and mesosphere, followed by $\alpha = 1.0$ and $\alpha = 0.0$. In the TR and SH, the $\alpha = 1.0$ results are worse than $\alpha = 0.0$ in the stratosphere.

The larger errors in $\alpha = 1.0$ may be related to spurious resolved gravity waves being generated in the system. To identify gravity waves, Fig. 16 shows global divergence maps on 1 December 2014 (00Z) at 10.5 hPa for the OAE with three blending coefficients, along with the zonal standard deviation of the divergence as a function of latitude. The divergence at this level clearly increases over the globe with more ensemble information added to the system. The globally averaged divergence profiles are provided in Fig. 17. These show increasing divergence with altitude, as expected for upward propagating gravity waves. Also, the divergence is enhanced at all vertical levels with larger value of $\alpha$. This suggests that local imbalance due to the use of localized ensemble covariance may be causing gravity waves that are propagating upward into the stratosphere and mesosphere (see Keypert, 2009 and Allen et al., 2015 for discussions of imbalance in the framework of the shallow water model and EnKF). Although more work is necessary to sort out the details, using $\alpha = 0.5$ likely provides the best results by combining reduced spurious imbalance relative to $\alpha = 1.0$ as well as enhanced flow-of-the-day information relative to $\alpha = 0.0$. We will use $\alpha = 0.5$ as the blending coefficient for the following sensitivity tests as well as for the combined ozone/radiance assimilation experiments in Sect. 5.

**4.3 Dependence on sampling pattern**

The previous results show that with global hourly coverage, ozone observations are able to constrain the stratosphere to wind error of less than about 2 ms$^{-1}$ and $T$ errors less than about 1 K. These are approximately the limits indicated by the bounds of the null hypothesis set examined in Sect. 4.1, indicating that the results are near the discernable limit for this system. The global sampling is, of course, unrealistic in both horizontal and temporal coverage. Here we examine sensitivity to sampling by repeating the $\alpha = 0.5$ experiments with polar-orbiting and random sampling (see Fig. 1b,c). The polar-orbiting sampling would be similar, for example, to the MLS or Ozone Mapping and Profiler Suite (OMPS). The random sampling is not realistic,

but provides a hypothetical test of what would happen if random observations occurred with the same frequency as the polar orbiter. In each case, we assume perfect ozone observations and $\sigma_{ob}$ of 0.1 ppmv. The RMS error profiles, averaged over the last 6 days of the experiments, are provided in Fig. 18.

The OAE error profiles for vector wind show that assimilation of both polar-orbiting (blue) and random (red) perfect ozone observations reduce the errors relative to the BE (dotted line). Particularly in the lower stratosphere, from about 100 to 10 hPa, the wind errors are relatively small, less than about 4 ms$^{-1}$. In the upper stratosphere (above 10 hPa), the errors for the polar-orbiting observations increase sharply with altitude to ~50 ms$^{-1}$ at 1.0 hPa. The wind errors for random sampling are consistently lower than for polar-orbiting, even though both contain approximately the same number of observations. While
there may be some redundancy in the polar-orbiting observations due to closely spaced along-track profiles, it is also likely that the large gaps between orbit tracks (see Fig. 1b) make it difficult for the polar-orbiting observations to completely constrain the winds. While the random sampling does better than polar-orbiting, there are still rather large wind errors in the random sampling, up to ~25 m/s in the NH and ~10 m/s in the TR. We note that error reductions occur even in the mesosphere, where there are no observations, suggesting that improving the stratospheric analyses also improves the mesosphere.

The OAE error profiles for $T$ also show improvements relative to the BE when polar-orbiting observations are assimilated, with the smallest errors occurring in the lower stratosphere. However, in the NH upper stratosphere the polar-orbiting observations only constrain $T$ to ~15 K. The experiment with random observations has smaller $T$ errors, which are similar to the global errors in the NH and TR up to about 10 hPa. In the SH lower stratosphere, the polar-orbiting and random cases
actually have smaller $T$ errors than the global case, and the global case has errors larger than the BE. However, the magnitude of these errors are near the error limit discussed in Sect. 4.1. Overall, we see that the ozone-dynamical influence is strongly sensitive to the sampling pattern, but wind and $T$ improvements are possible even with a realistic polar-orbiting satellite.

## 4.4 Dependence on observation error

Next, we examine the sensitivity of the analysis to the ozone observation error. First, we assimilate polar-orbiting data with
2% error (green solid lines on Fig. 18). This is a realistic error value for the middle stratosphere; for example, Aura MLS V4.2 precision specifications are rated at 2% at 22, 10, and 5 hPa and greater than 2% elsewhere (Livesy et al., 2016). The results show slightly larger vector wind and $T$ errors in the NH for 2% error than when perfect data were assimilated, but the errors are less than the background throughout the stratosphere and mesosphere, suggesting value added by these observations. In the tropics and SH, the 2% case is also very similar to the 0% case for both vector wind and $T$. These results suggest that
assimilating actual profile measurements with realistic errors can potentially benefit the analyses.

We now add random noise to the global observations using Gaussian errors of 2%, 5%, and 10% to further examine sensitivity to errors. For each of these three cases, the specified $\sigma_{ob}$ is also set to the same percent value, but with a lower limit of 0.1 ppmv (see Table 1, Sect. 4.4). The results, in Fig. 19, show that adding noise increases the vector wind errors over the perfect observations. Below about 10 hPa, the wind errors are similar for 2%, 5%, and 10% cases, while above 10 hPa, there is generally increased error with increased observational noise. The stratospheric wind errors are still relatively small (less than ~10 ms$^{-1}$, even with 10% error, suggesting that the dynamic variability of ozone is large enough to allow wind information on this error level.

The OAE $T$ errors (Fig. 19d,e,f) are also generally larger with more observational noise. With 10% applied ozone errors, the stratospheric $T$ errors are constrained to within ~6 K in the NH, ~2K and TR, and ~1 K in the SH. In the SH lower stratosphere, there is a reversal of the $T$ errors, with the 10% case showing smaller errors than the 5% or 2% cases. The cause of this reversal is uncertain, but it may be that using higher $\sigma_{ob}$ in the 10% case reduces the weight of the observations and therefore results in reduced spurious errors relative to the TE. Overall, we conclude here that noisy observations will generally reduce the amount of wind information that can be derived from ozone. As a caveat, we remind the reader that we are only simulating random error and not biases, which could be a significant source of additional error.

## 5. Ozone and radiance assimilation

### 5.1 Baseline experiment for radiance assimilation

We next examine the impact of ozone when the stratosphere is already constrained by radiance observations. As described in Sect. 2.3, we simulated infrared and microwave radiance observations for AMSU-A, AIRS, ATMS, and IASI for the $\alpha = 0.5$ TE case, and then added random noise. The vector wind and $T$ error profiles for radiance-only experiments are provided in Fig. 20. The grey lines shows the results for assimilation of noisy radiances. In the stratosphere, wind errors range from around 2 to 4 ms$^{-1}$, while $T$ errors range from around 0.5 to 1.5 K. These are relatively small errors, making error reduction via ozone assimilation more challenging than when only ozone was assimilated in the stratosphere. Comparing the noisy radiance results in Fig. 20 with the noisy global ozone (2% error) from Fig. 19, we see that ozone assimilation has smaller wind errors throughout most of the stratosphere in all three latitude bands. For $T$, however, the radiance assimilation has generally smaller errors in the extratropics, while $T$ errors are similar for ozone or radiances in the tropics.

Before combining ozone with radiances, we also performed an experiment in which "perfect" radiances were assimilated with unperturbed initial errors, as we did with ozone in Sect. 4.1. Wind and $T$ error profiles from this case are shown with black lines on Fig. 20. The experiment with perfect radiances results in errors in both the troposphere and stratosphere that are slightly larger than the case of perfect ozone with unperturbed initial conditions (Fig. 14). Potential reasons for the difference

may include the interaction of deep vertical weighting functions of radiances observations with the tropospheric state or the much larger number of radiance observations compared to ozone observations. We expect the errors for the combined ozone/radiance experiments to lie within the grey and black lines of Fig. 20.

## 5.2 Ozone and radiance assimilation experiments

In the next set of experiments, ozone data (global, random, and polar-orbiting) are assimilated along with the noisy radiance observations. Figure 21 shows vertical profiles of the resulting errors for perfect ozone observations for the three sampling patterns. We also include in Fig. 21 the error profiles (black and grey lines) from the radiance assimilation experiments shown in Fig. 20 as a comparison. In the TR, the impact of ozone assimilation is positive for all three sampling patterns, with generally increasing impact with altitude throughout the stratosphere. Global observations reduce tropical vector wind errors by up to ~2 ms$^{-1}$ at the stratopause, while random and polar-orbiting data reduce tropical wind errors by about 1.0 and 0.4 ms$^{-1}$, respectively. In the NH and SH, global observations benefit winds throughout the stratosphere, but at a reduced amount compared with the tropics. The impact of random observations in the NH and SH is positive throughout the stratosphere, but at smaller levels (~0.1 to 0.3 ms$^{-1}$) than for global data. For polar-orbiting observations, the impact on NH and SH winds is even smaller, but still generally positive. We note that the error profiles generally lie within the perfect and noisy radiance profiles, and the vector wind errors for global ozone is close to the perfect radiance profile. This suggests that the ozone is reducing wind errors to near the minimum possible values, identified by the perfect radiance case.

Temperature error reductions show a similar pattern to the wind errors, with largest impact from the global observations in the tropical upper stratosphere, where reductions of ~0.7 K occur. The random and polar-orbiting observations also impact the tropical upper stratosphere, with $T$ error reductions of ~0.3 and ~0.1 K, respectively. In the extratropics, the impact of the random and polar-orbiting observations on $T$ is small, generally less than ~0.1 K. We also note that impact of ozone observations on both winds and $T$ is generally positive in the mesosphere, above the highest observation level. Global ozone assimilation results in temperature errors that are slightly lower (higher) than the perfect radiance case in the tropics (NH and SH). In terms of vertically-averaged errors (over the range of ozone observations, 77 - 1 hPa), the overall impact on both winds and $T$ is positive throughout the time period of consideration (see Fig. 13a,b). While six days is likely too small for statistical significance tests, the pattern of error reductions is consistent, with polar data providing a slight improvement, followed by random data. The errors for global data are similar to the perfect radiance error case.

For further quantification of ozone impact on the dynamics, we also calculated the error difference (OAE errors minus the noisy radiance errors), where negative values of this difference indicate value added due to ozone assimilation. Figure 22 plots these quantities as zonal mean cross-sections. For global ozone, the vector winds and $T$ are improved throughout much of the stratosphere and lower mesosphere, with peak reductions of ~3 ms$^{-1}$ and ~1 K in the tropical upper stratosphere. The random and polar sampling patterns also improved the tropical winds and temperature, but to a lesser extent than the global data. One

reason for the maximum impact is in the tropics is due to the dynamics being less constrained by radiance measurements due to lack of geostrophy. As demonstrated by Allen et al. (2015), using a shallow water model with simulated ozone and height observations, ozone assimilation particularly improved the tropical winds. In addition, as discussed by Daley (1996), the impact on winds from tracer assimilation also depends on tracer time tendency. When this tendency is small, wind recovery will occur slowly or not at all. Small tendency can occur when the tracer gradient is small, when the wind speed is low, or when the tracer and streamfunction are highly correlated. To test this, we averaged the absolute value of the 1-h time tendency over the last 6 days of the TE at each grid point and then calculated the zonal mean. The results (Fig. 23) show the strongest tendencies in the tropical lower and upper stratospheres, in the NH polar upper troposphere/lower stratosphere, and in the SH lower stratosphere. The tropical maxima roughly coincide with regions of large wind and $T$ error reductions in Fig. 22, but there are no corresponding error reductions in the extratropical lower stratospheres. This may be due to the radiances having a large influence there with many observations and strong geostrophic coupling, making it difficult for ozone observations to add value.

Next, in Figs. 24 and 25, we repeat the above comparison with noisy ozone data (2% applied error). The wind error profiles (Fig. 24) show much less variation from the noisy radiance profile. Global ozone benefits the winds over a large altitude range and also benefits $T$, particularly in the tropical upper stratosphere. It is difficult to discern impacts of random and polar ozone from these plots, except in the lower mesosphere, where there is a slight benefit. This is also seen in the vertically integrated error time series (Fig. 13c,d), where the global ozone is consistently smaller than the noisy radiance alone. The random and polar errors are not easily distinguishable from noisy radiance. It is not surprising that the radiances overwhelm these noisy polar and random ozone observations since the total number of ozone observations (3500/4 profiles x 17 observations/profile = 14,875) is only ~1% of the number of radiance observations (e.g., 1,768,409 observations shown in Fig. 2) for a given 6-h cycle.

The zonal mean error differences for the 2% ozone error case are provided in Fig. 25. Wind and $T$ error reductions are seen for the global ozone in the tropical upper stratosphere and lower mesosphere, while the random and polar only show strong influence in the tropical lower mesosphere. Slight increases in $T$ errors also occurs for global ozone in the NH polar region. Whether these error impacts are statistically significant cannot be determined from the limited number of samples. However, a general pattern of improvement from ozone assimilation in the tropical stratosphere and mesosphere emerges from these experiments, providing encouragement for future work with more realistic observing systems, model resolution, and extended analyses to allow significance testing.

## 6. Summary and Conclusions

This study examined the potential impact of stratospheric ozone assimilation on the wind and temperature analyses in the stratosphere and lower mesosphere for a case study in late November/early December 2014. We used unbiased simulated measurements and a perfect model to test the ozone-dynamics interaction in hybrid 4D-Var DA. The structures of the ensemble cross-correlations for ozone with other variables were illustrated with a composite single ozone observation increment, formed by averaging the spatial cross-correlations for 36 points around a latitude circle. Clear patterns emerged that included rotation around a vertical axis and a vertical temperature dipole. These patterns resembled the potential vorticity "charge" concept, discussed by Bishop and Thorpe (1994). This suggests that an ozone observation, at least in the presence of sufficient spatial gradients and geostrophic balance, acts like an observation of potential vorticity. This is likely due to both quantities being quasi-conserved in the stratosphere and therefore forming compact relationships. Further work on the understanding of these relationships may provide insight into designing ozone-observing systems that would optimize the ozone-wind relationship.

Experiments were then conducted in which simulated stratospheric observations were assimilated in a cycling hybrid 4D-Var system. The resulting analyses were compared with a truth experiment that was used for simulating the observations and verifying the analyses. All experiments included a suite of conventional observations to constrain the troposphere. This approach allowed a controlled method for determining ozone impact on the stratospheric dynamics, while maintaining a realistic troposphere. Experiments assimilated various combinations of stratospheric ozone and radiances. The mechanisms through which ozone can impact the winds in hybrid 4D-Var include both the application of cross-covariances of ozone with other fields in the initial blended background error covariance and the use of the ozone continuity equation in the tangent linear model/adjoint. We showed that using a blending coefficient of 0.5 provided better results than either 0.0 or 1.0. This is likely due the combined positive effects of the ensemble flow-of-the-day information with the negative aspects of spurious unbalanced modes spawned by the localized ensemble covariance. These aspects were discussed in the shallow water model context in Allen et al. (2016), where it was shown that the optimal blending coefficient also depends both on the data being assimilated and on the ensemble size.

Ozone assimilation can benefit the winds and temperature if sufficient high-quality observations are available. For example, global hourly ozone data with no error constrained the stratosphere to within a few ms$^{-1}$ for the winds and ~1 K for temperature, which was better than noisy radiance assimilation, but worse than perfect radiance assimilation. When ozone is assimilated with noisy radiances, wind improvement is mostly found in the tropics, where the lack of geostrophic balance renders radiances less effective. For example, assimilating realistic radiance data and perfect global ozone data resulted in additional tropical wind and temperature error decreases in the upper stratosphere of ~3 ms$^{-1}$ and ~1 K relative to the noisy radiance data. However, when a realistic 2% error was added to the global ozone data, the tropical error decreases were reduced to ~1 ms$^{-1}$

and ~0.5 K, and slight error increases occurred in the NH polar region. Reduction of the sampling frequency also reduced the benefit of ozone.

We are unable to establish the statistical significance of the additional error reductions for the cases of noisy ozone assimilation,
since much longer assimilation tests are required to establish significance for changes of less than a percent (Geer, 2016). When looking for very small improvements from ozone assimilation, several other factors are likely to become important. We limited the study to unbiased ozone and radiance observations. Further work is necessary to determine the impact of ozone assimilation in a system with model and/or observation biases. Also, our experiments simulated ozone measurements on the model vertical grid. For lower vertical resolution ozone measurements such as Solar Backscatter Ultraviolet or OMPS Nadir
Profiler (Flynn et al., 2009, 2014), or ozone sensitive channels of infrared sounders, the observation operator must include a vertical weighting function. Further studies are required to determine the ozone vertical resolution requirements for achieving wind improvements. Similarly, unlike this study, the horizontal resolution of current NWP systems is higher than the resolution of limb sounding ozone measurements such as MLS (Livesy et al., 2015) and OMPS Limb Profiler (Jaross et al., 2014). This requires the use of horizontal weighting functions in the observation operator for optimal assimilation. All these issues will be
important for achieving wind improvements from ozone assimilation with current NWP systems and ozone measurement technology.

In this study, we only simulated ozone vertical profile measurements, since we expect that vertical resolution is essential for the ozone-wind relationship to be robust. Total column observations from Ozone Monitoring Instrument (OMI) or OMPS
could provide supplementary information to constrain the winds (see, for example, the study by Peuch et al., 2000). Certain radiance channels also have ozone sensitivity that could potentially be exploited (Dragani and McNally, 2013). Other approaches to the ozone-dynamical impact could include assimilation of ozone radiances directly into the system rather than retrieved profiles. In addition, the impact of assimilation of other tracers could be tested in a similar framework. For example, Andersson et al. (1994, 2007) have shown dynamical impacts from assimilation of radiance channels sensitive to water vapor,
and Allen et al. (2014), in a shallow water model study, showed that nitrous oxide could additionally improve winds in 4D-Var DA. Given the potential benefit of tracer assimilation on the dynamics in NWP (referred to by Daley (1995) as a "tantalizing possibility"), it is our hope that this study will motivate future work in this area and eventually result in improved operational analyses and forecast skill.

**Acknowledgments**

We would like to thank Alan Geer, Thomas Milewski, and the Co-Editor (William Lahoz) for helpful comments on this manuscript. This work was funded by the U. S. Office of Naval Research. Douglas R. Allen and Karl W. Hoppel acknowledge

support from Office of Naval Research base funding via Task BE-033-02-42. David D. Kuhl and Karl W. Hoppel acknowledge support from Office of Naval Research base funding via Task BE-435-050.

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

| Experiment Type | α | Ozone Observation Sampling | Ozone Error [%] | $\sigma_{ob}$ [constant or %/minimum] | Radiance Observations | Radiance Error | Initial Conditions |
|---|---|---|---|---|---|---|---|
| *Sect. 4.1* | | | | | | | |
| Truth | 0.5 | | | | | | Unperturbed |
| Ozone Only | 0.5 | global | 0 | 0.1 ppmv | | | Unperturbed |
| Ozone Only | 0.5 | random | 0 | 0.1 ppmv | | | Unperturbed |
| Ozone Only | 0.5 | polar-orbiting | 0 | 0.1 ppmv | | | Unperturbed |
| *Sect. 4.2* | | | | | | | |
| Truth | 0.0 | | | | | | Unperturbed |
| Truth | 0.5 | | | | | | Unperturbed |
| Truth | 1.0 | | | | | | Unperturbed |
| Baseline | 0.0 | global | 0 | 0.1 ppmv | | | Perturbed |
| Baseline | 0.5 | global | 0 | 0.1 ppmv | | | Perturbed |
| Baseline | 1.0 | global | 0 | 0.1 ppmv | | | Perturbed |
| Ozone Only | 0.0 | global | 0 | 0.1 ppmv | | | Perturbed |
| Ozone Only | 0.5 | global | 0 | 0.1 ppmv | | | Perturbed |
| Ozone Only | 1.0 | global | 0 | 0.1 ppmv | | | Perturbed |
| *Sect. 4.3* | | | | | | | |
| Ozone Only | 0.0 | global | 0 | 0.1 ppmv | | | Perturbed |
| Ozone Only | 0.5 | random | 0 | 0.1 ppmv | | | Perturbed |
| Ozone Only | 0.5 | polar-orbiting | 0 | 0.1 ppmv | | | Perturbed |
| *Sect. 4.4* | | | | | | | |
| Ozone Only | 0.5 | global | 0 | 0.1 ppmv | | | Perturbed |
| Ozone Only | 0.5 | polar-orbiting | 2 | 2%/0.1 ppmv | | | Perturbed |
| Ozone Only | 0.5 | global | 2 | 2%/0.1 ppmv | | | Perturbed |
| Ozone Only | 0.5 | global | 5 | 5%/0.1 ppmv | | | Perturbed |
| Ozone Only | 0.5 | global | 10 | 10%/0.1 ppmv | | | Perturbed |
| *Sect. 5.1* | | | | | | | |
| Baseline | 0.5 | | | | All | Noisy | Perturbed |
| Baseline | 0.5 | | | | All | Perfect | Unperturbed |
| *Sect. 5.2* | | | | | | | |
| Radiance & Ozone | 0.5 | global | 0 | 0.1 ppmv | All | Noisy | Perturbed |
| Radiance & Ozone | 0.5 | random | 0 | 0.1 ppmv | All | Noisy | Perturbed |
| Radiance & Ozone | 0.5 | polar-orbiting | 0 | 0.1 ppmv | All | Noisy | Perturbed |
| Radiance & Ozone | 0.5 | global | 2 | 2%/0.1 ppmv | All | Noisy | Perturbed |
| Radiance & Ozone | 0.5 | random | 2 | 2%/0.1 ppmv | All | Noisy | Perturbed |
| Radiance & Ozone | 0.5 | polar-orbiting | 2 | 2%/0.1 ppmv | All | Noisy | Perturbed |
| Radiance & Ozone | 0.5 | global | 2 | 2%/0.1 ppmv | All | Noisy | Perturbed |

**Table 1.** Experiment descriptions used in each subsection of this study. Columns indicate (1) experiment type, (2) covariance blending value, (3) ozone observation sampling, (4) ozone observation error, (5) background ozone error standard

deviation (given as constant mixing ratio in ppmv or % value with minimum threshold in ppmv), (6) radiance observations, (7) radiance error, and (8) initial conditions.

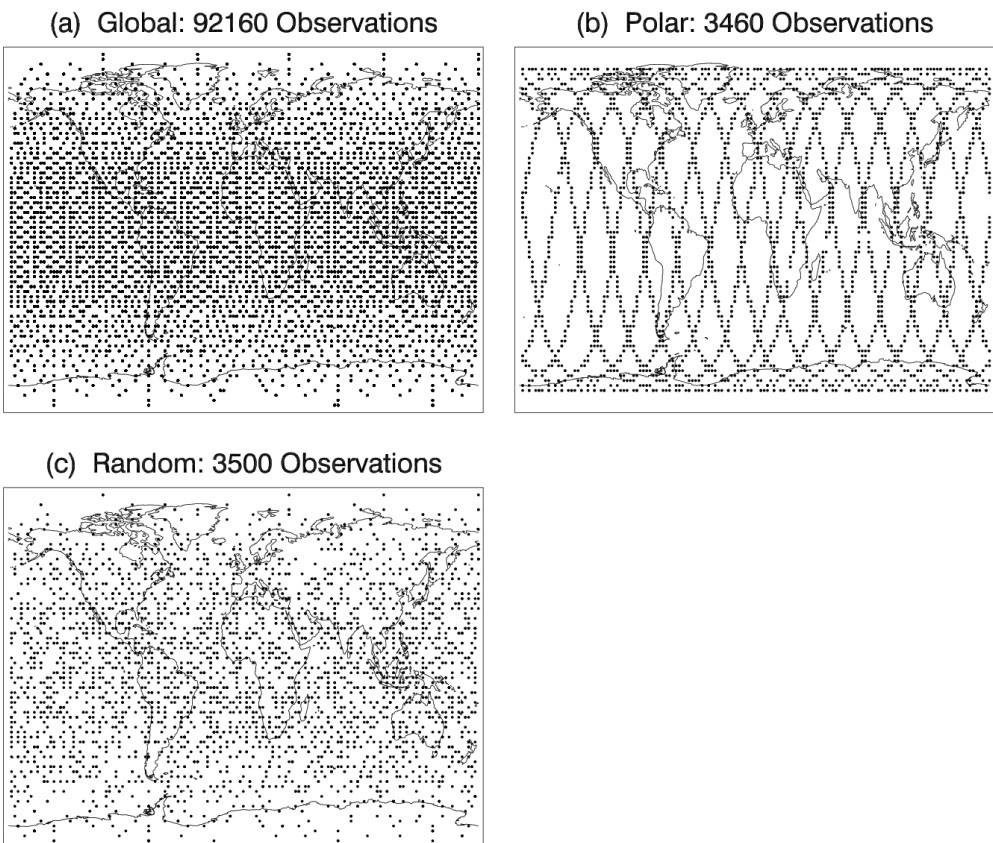

**Fig. 1.** Locations of simulated ozone observations for (a) global, (b) polar-orbiting, and (c) random data. For global data, these represent hourly coverage, while for polar and random data these are all the observations over a 24-h period.

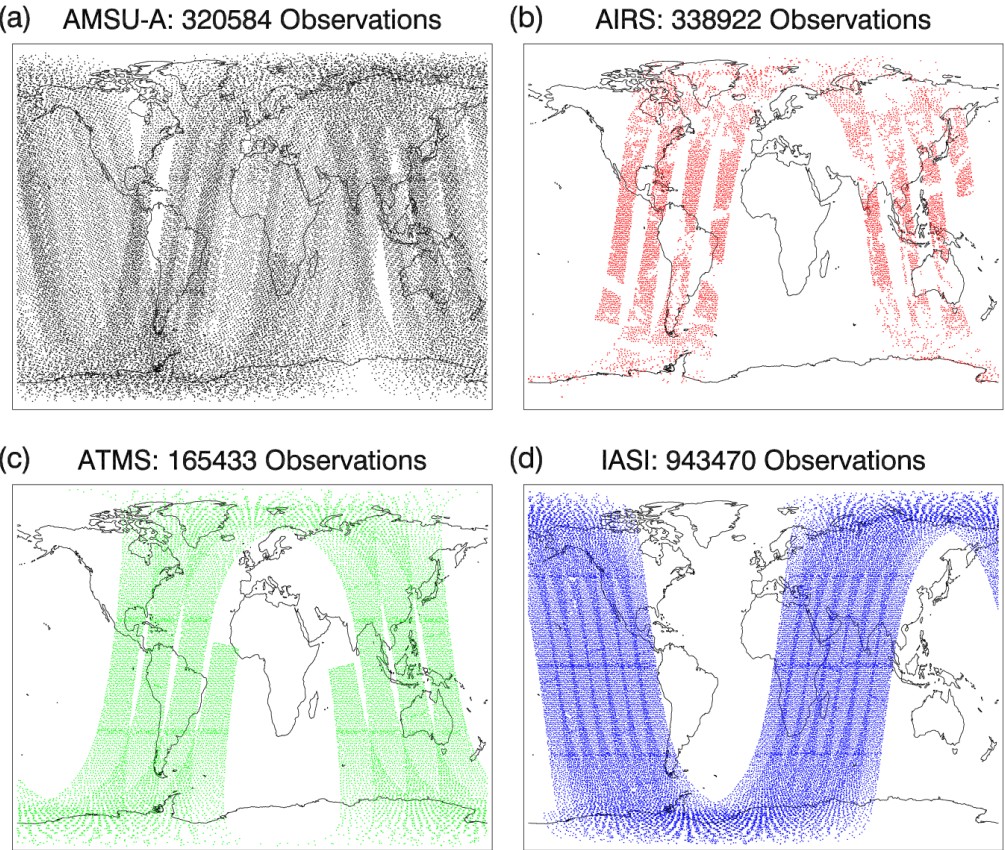

**Fig. 2.** Locations of simulated radiance observations for one 6-h update cycle centered on 15 November 2014 (06Z). Panels show observation locations for (a) AMSU-A, (b) AIRS, (c) ATMS, and (d) IASI.

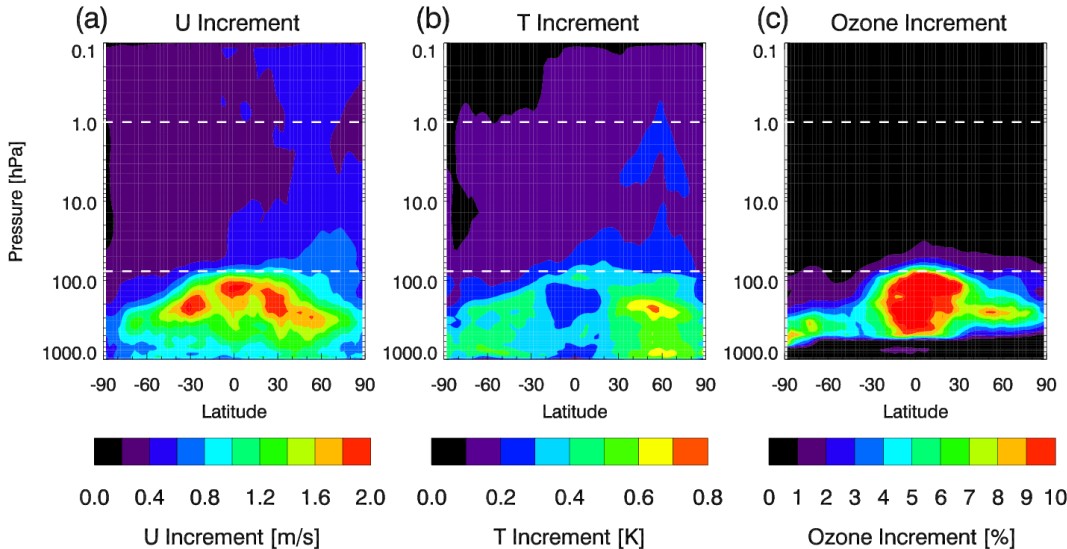

**Fig. 3.** Zonal mean (pressure vs. latitude) plots of (a) the absolute value of the *u* increment [in ms⁻¹], (b) the absolute value of the *T* increment [in K], and (c) the absolute value of the ozone increment divided by the analyzed ozone [in %] for the truth experiment, averaged over all cycles from 25 November - 1 December 2014. Red (blue) indicating high (low) values. Region between the white dashed lines indicates vertical range of assimilated stratospheric ozone observations.

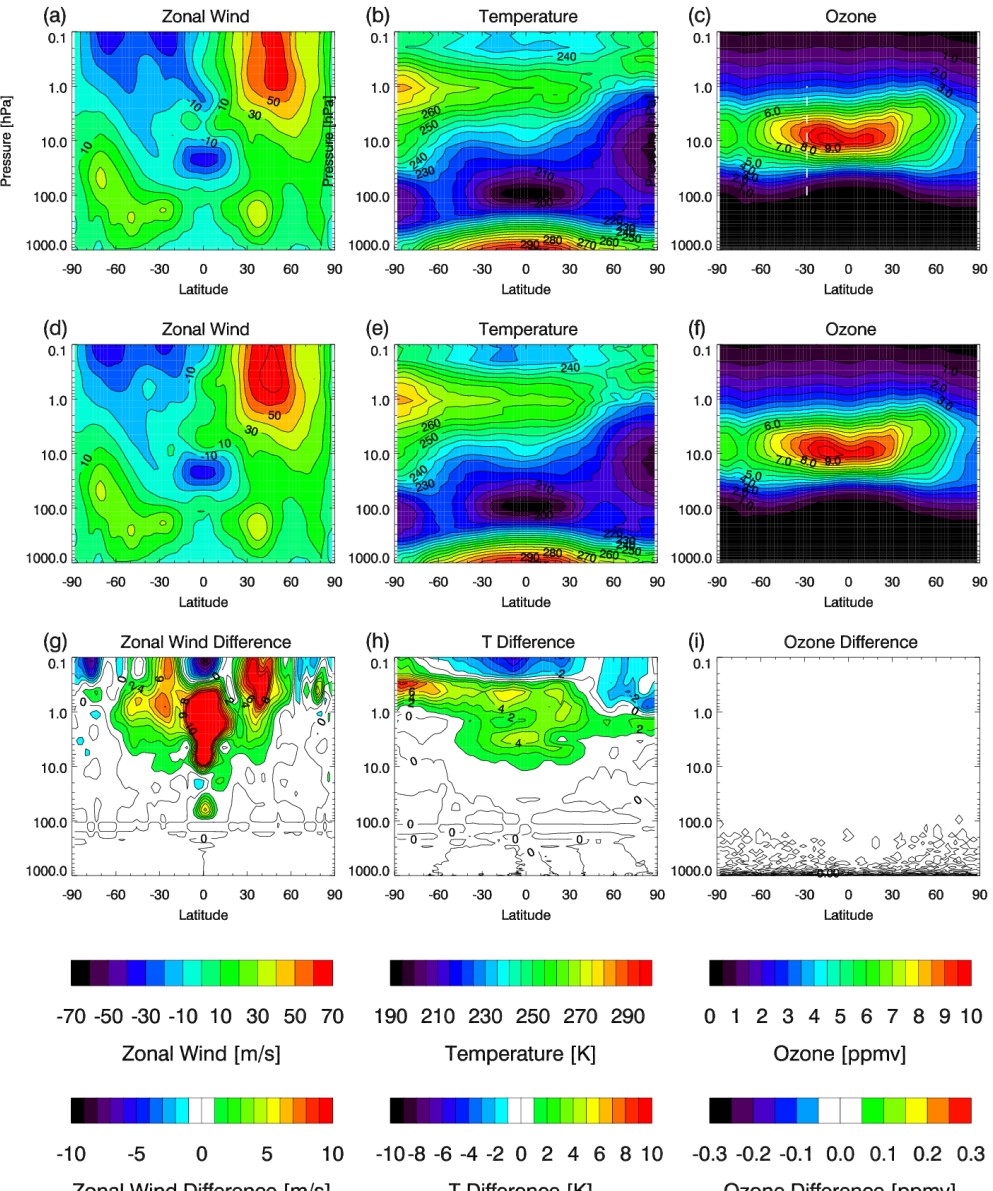

**Fig. 4.** Analyses for 15 November 2014 (00Z). Zonal mean zonal wind [ms$^{-1}$] for the (a) truth experiment and (d) baseline experiment and (g) baseline-truth zonal wind difference. Zonal mean temperature [K] for the (b) truth experiment and (e) baseline experiment and (h) baseline-truth temperature difference. Zonal mean ozone [ppmv] for the (c) truth experiment and (f) baseline experiment and (i) baseline-truth ozone difference. Color bars are provided for zonal mean quantities and differences. Red (blue) indicates high (low) values for each quantity. White dashed line in panel (c) indicates latitude of 28.6°S, which is the central latitude of the cross-correlations examined in Sect. 3.4

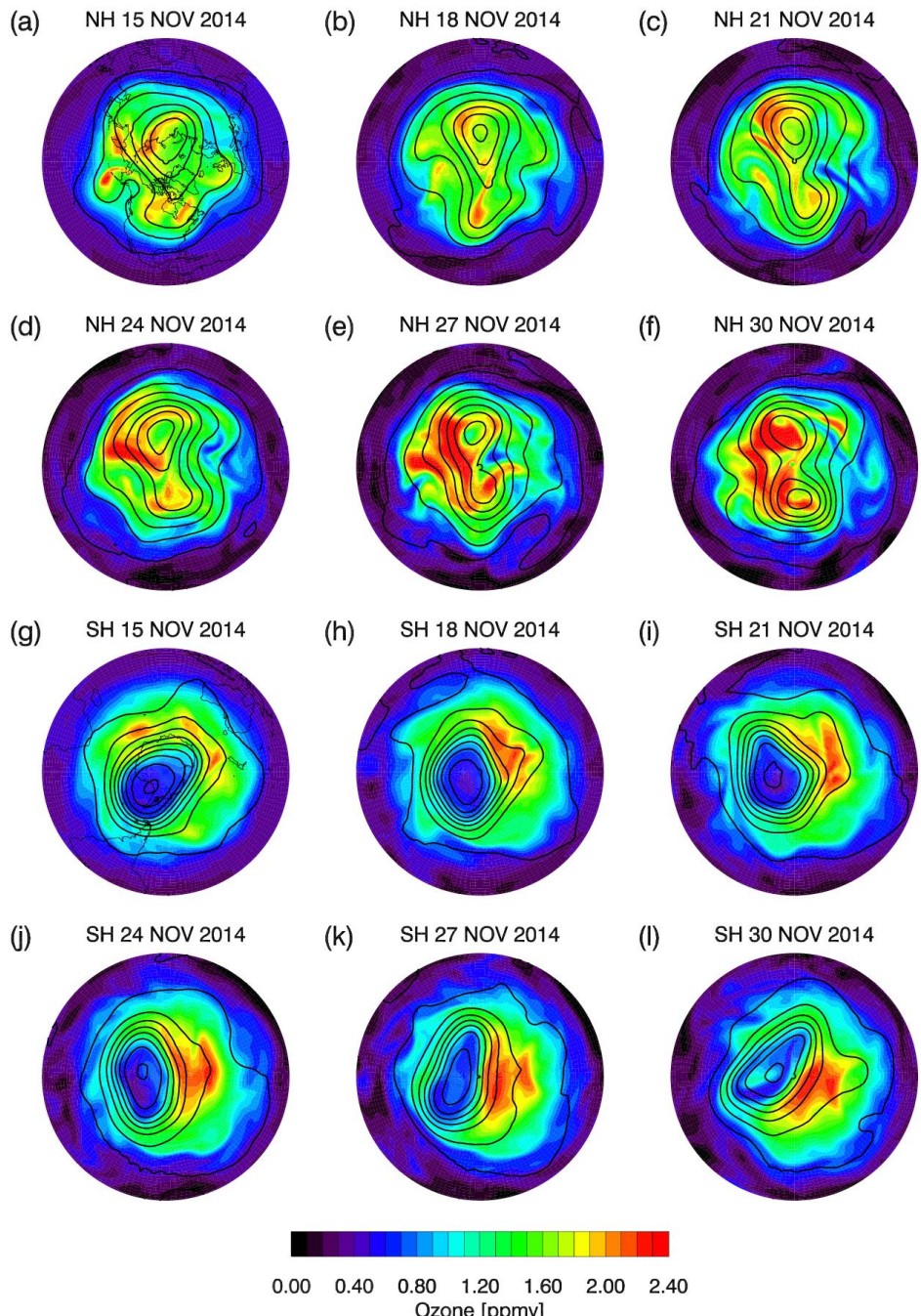

**Fig. 5.** Maps of ozone [ppmv] (colors) at 77 hPa (~18 km) overlaid with geopotential height (black lines) at 200 m intervals for 15, 18, 21, 24, 27, and 30 November 2014 (all at 00Z). (a-f) are NH and (g-l) are SH. Red (blue) contours indicated high (low) ozone values. Continent lines are placed on the maps for 15 November.

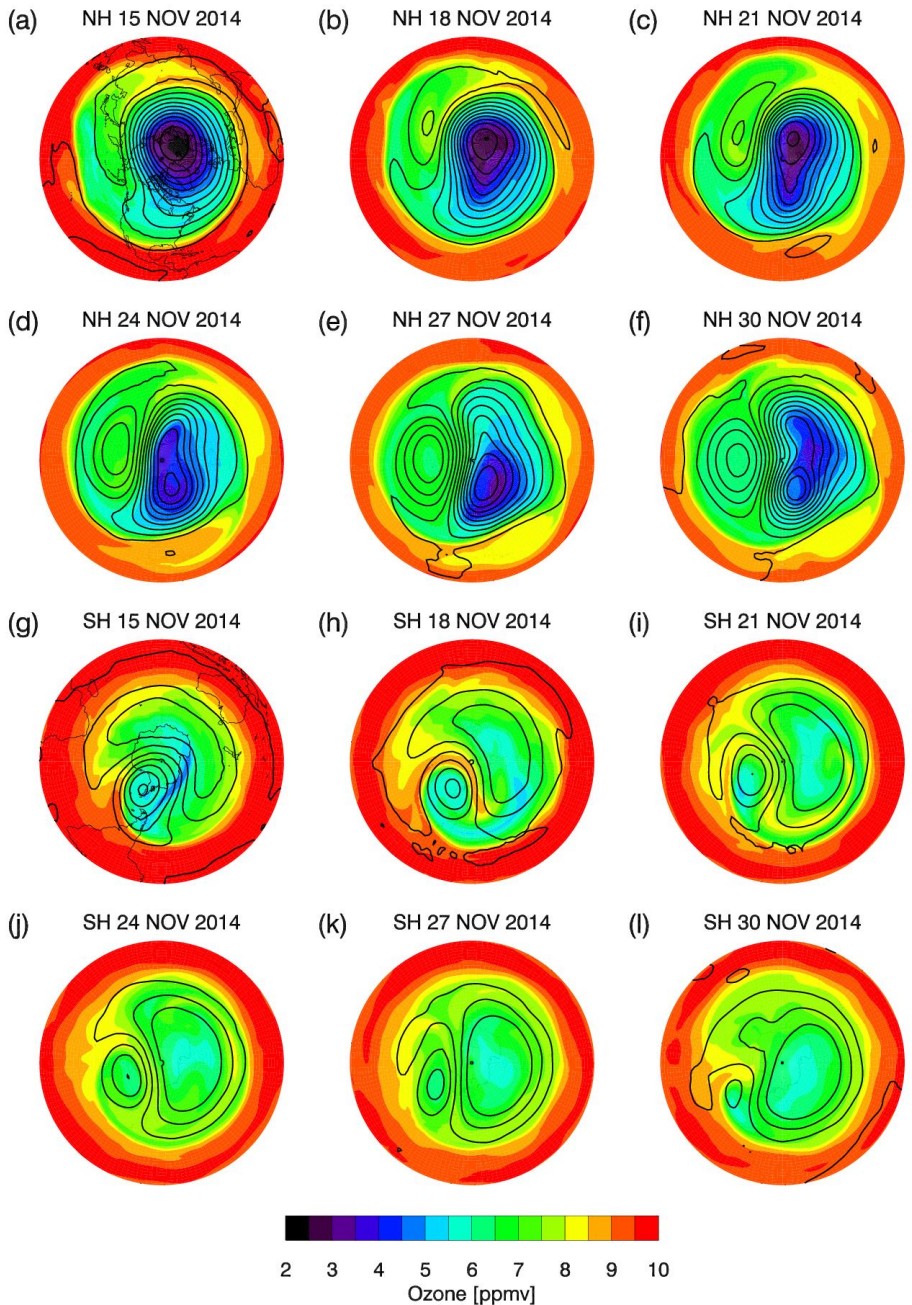

**Fig. 6.** Maps of ozone [ppmv] (colors) at 10.5 hPa (~32 km) overlaid with geopotential height (black lines) at 200 m intervals for 15, 18, 21, 24, 27, and 30 November 2014 (all at 00Z). (a-f) are NH and (g-l) are SH. Red (blue) contours indicated high (low) ozone values. Continent lines are placed on the maps for 15 November.

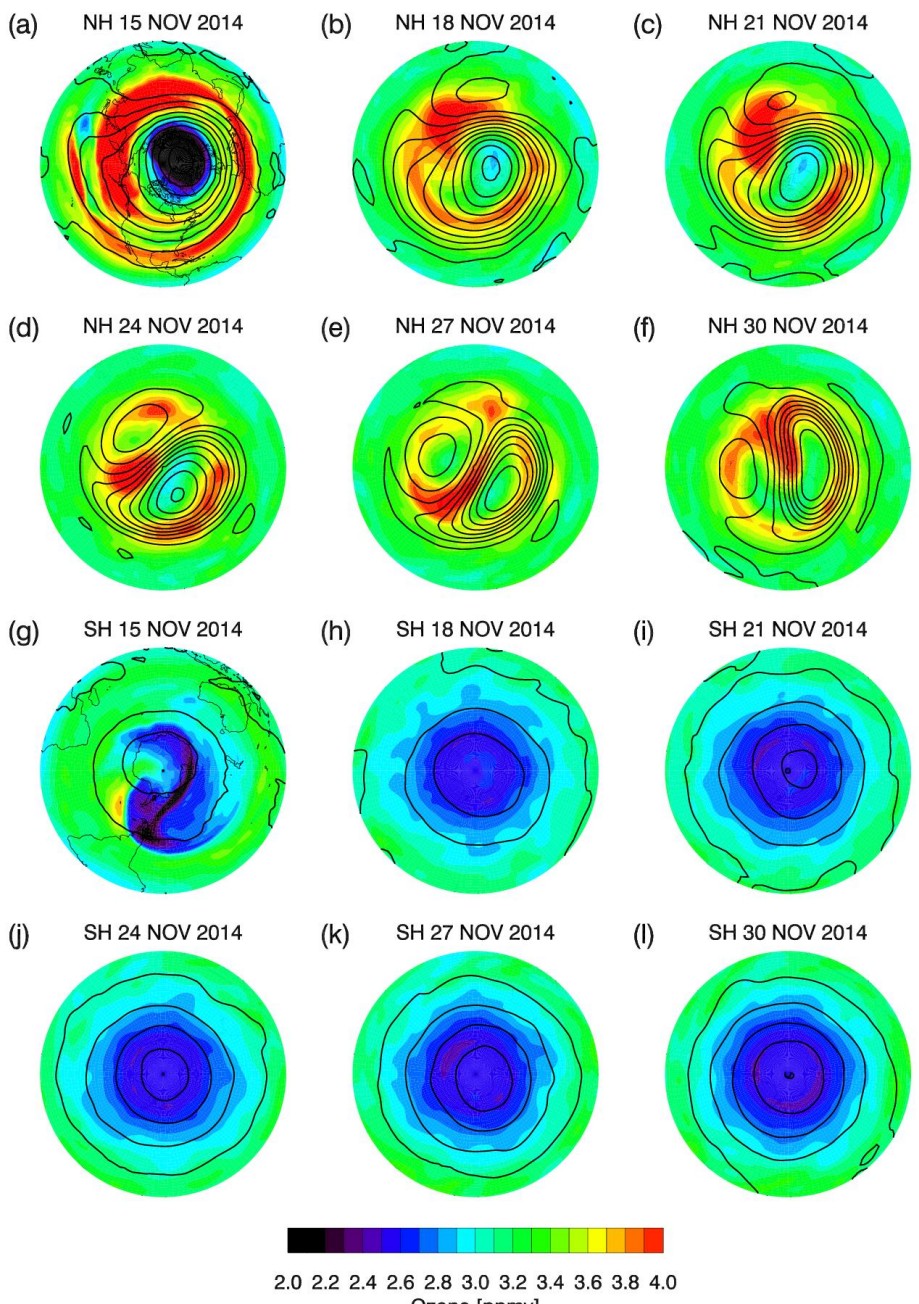

**Fig. 7.** Maps of ozone [ppmv] (colors) at 1.1 hPa (~48 km) overlaid with geopotential height (black lines) at 500 m intervals for 15, 18, 21, 24, 27, and 30 November 2014 (all at 00Z). (a-f) are NH and (g-l) are SH. Red (blue) contours indicated high (low) ozone values. Continent lines are placed on the maps for 15 November.

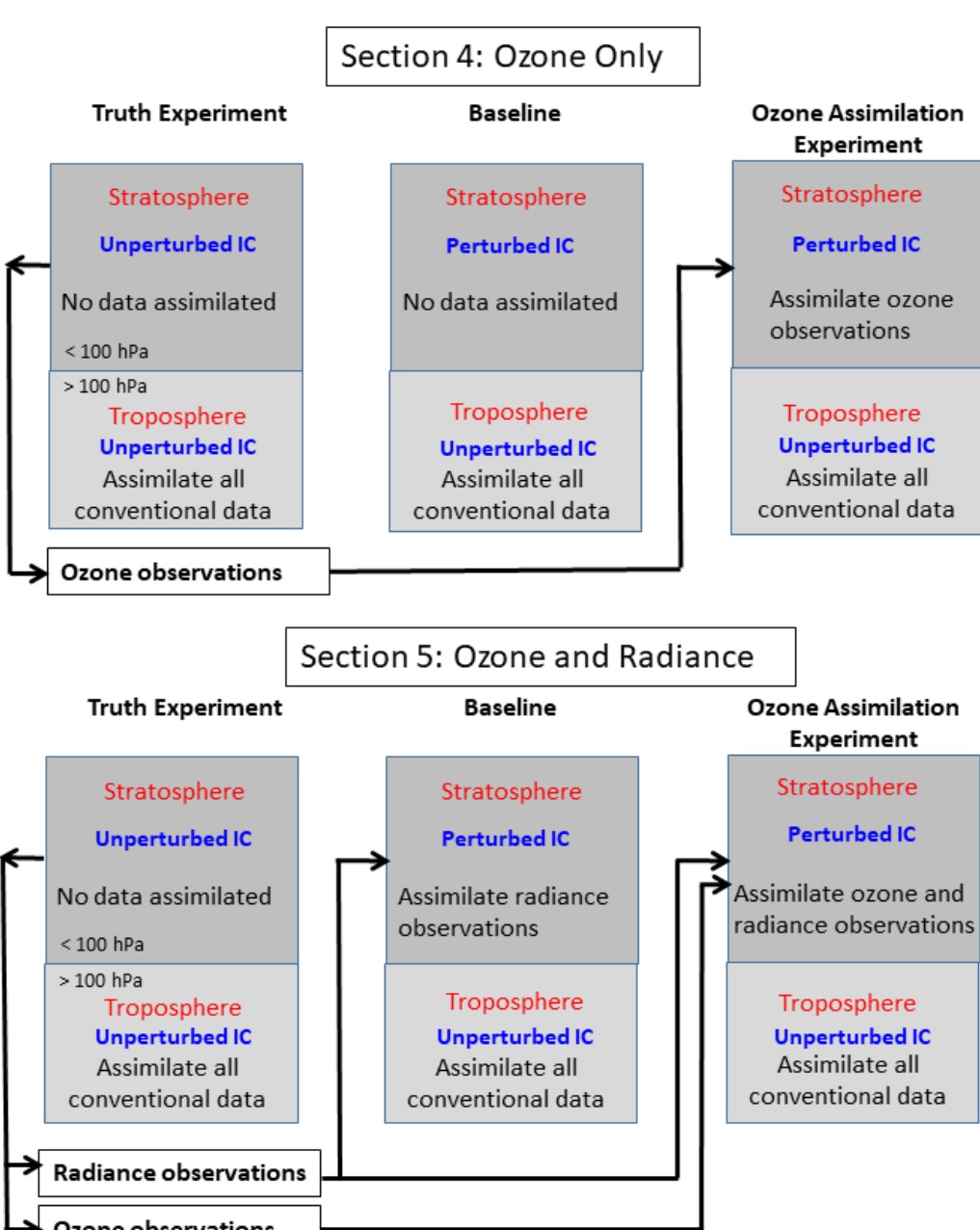

**Fig. 8.** Schematic diagram to illustrate the design for the experiments analyzed in Sect. 4 and Sect. 5. The truth experiment (TE) in both cases assimilates all conventional data in the troposphere (pressures greater than 100 hPa) and no data in the stratosphere (pressures less than 100 hPa). In Sect. 4, the baseline experiment is the same as the TE, except for perturbed

5     initial conditions (IC) in the stratosphere. The ozone assimilation experiment for Sect. 4 assimilates ozone observations created by the TE. In Sect. 5, the baseline includes assimilation of noisy radiance observations created from the TE. The ozone assimilation experiment in Sect. 5 includes both radiance and ozone observations from the TE.

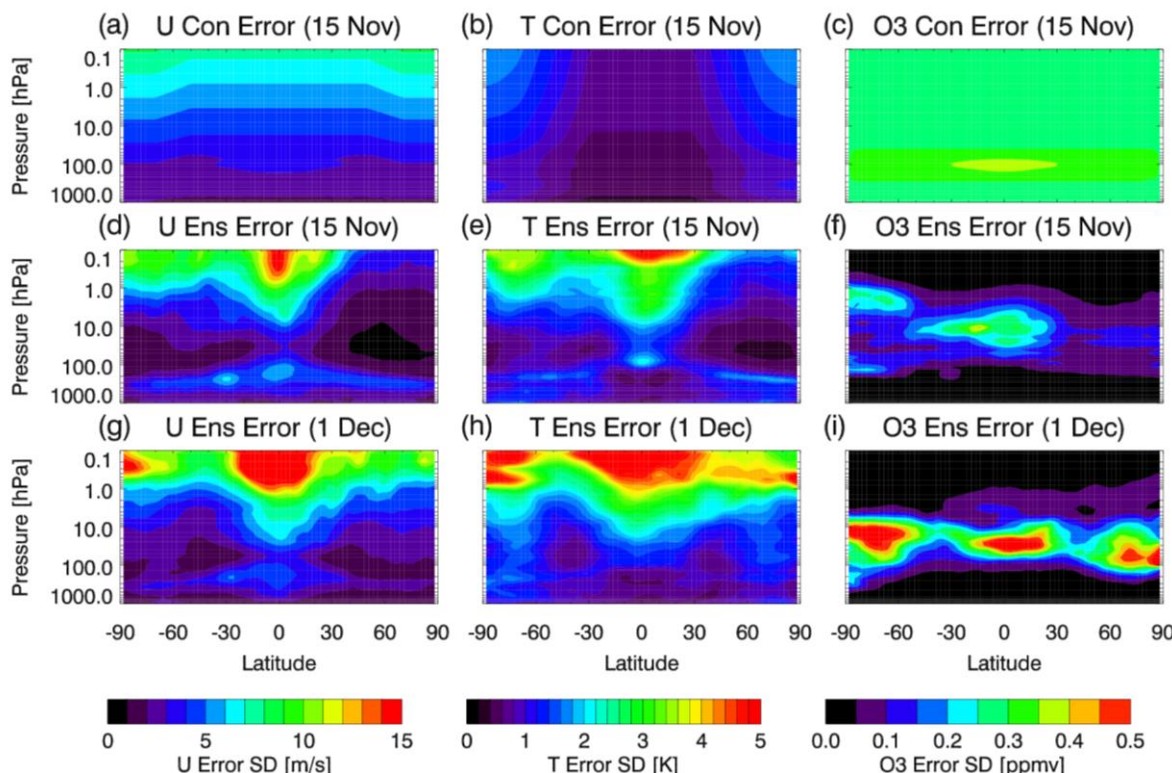

**Fig. 9.** Conventional background error standard deviations for (a) zonal wind [ms⁻¹], (b) temperature [K], and (c) ozone [ppmv] for 15 November 2014 (00Z). Ensemble background error standard deviations for (d) zonal wind, (e) temperature, and (f) ozone for 15 November 2014 (00Z). Ensemble background error standard deviations for (g) zonal wind, (h) temperature, and (i) ozone for 1 December 2014 (00Z). Color bars are provided for zonal wind, temperature, and ozone, with red (blue) indicating high (low) values.

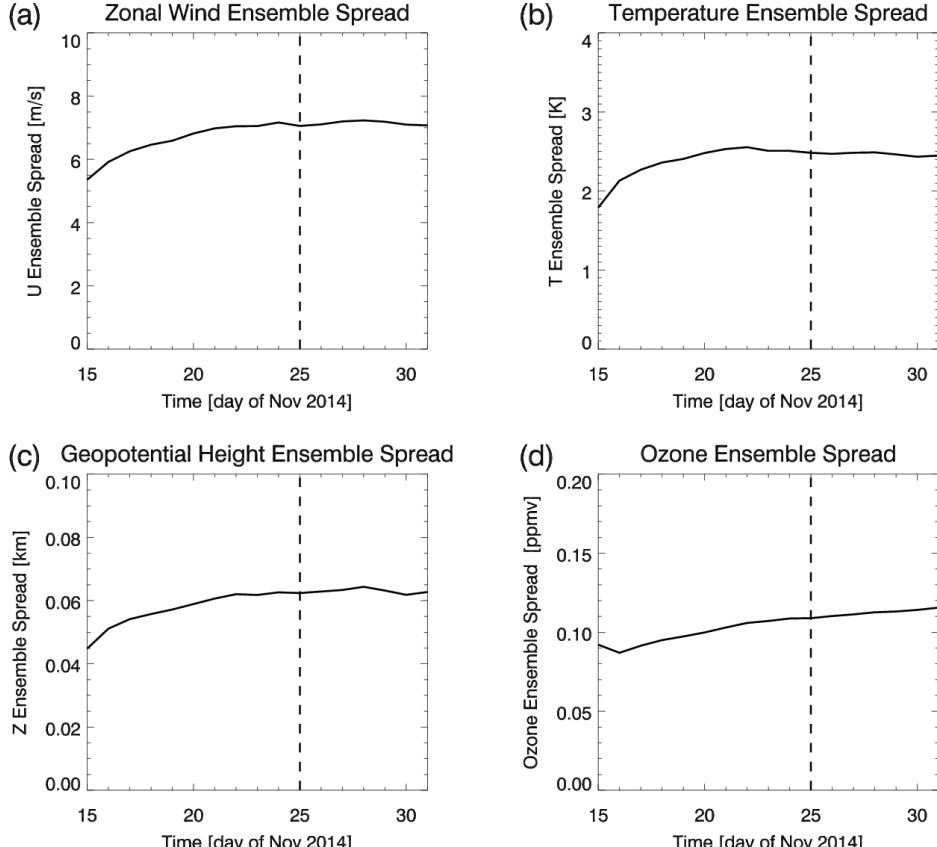

**Fig. 10.** Globally and vertically (from suface to model top) averaged (surface to model top) ensemble background error standard deviation (or "spread") for 15 November - 1 December 2014, evaluated at 00Z for each day. Panels are for (a) zonal wind [ms$^{-1}$], (b) temperature [K], (c) geopotential height [km], and (d) ozone [ppmv]. The dashed line on 25 November indicates the end of the spin-up period.

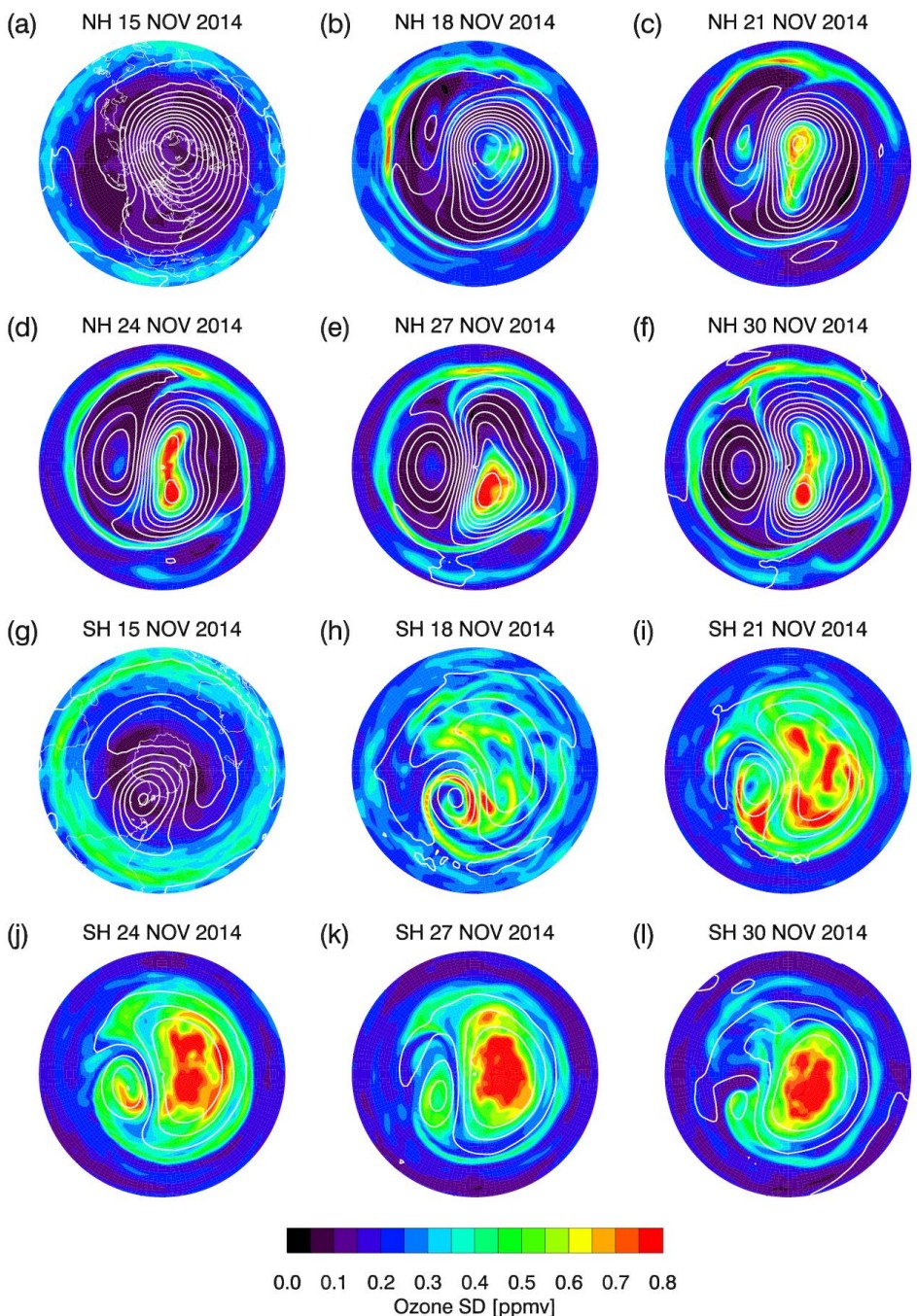

**Fig. 11.** Maps of ozone $\sigma_{ens}$ [ppmv] (colors) at 10.5 hPa (~32 km) overlaid with geopotential height (white lines) at 200 m intervals for 15, 18, 21, 24, 27, and 30 November 2014 (all at 00Z). (a-f) are NH and (g-l) are SH. Red (blue) contours indicated high (low) ozone values. Continent lines are placed on the maps for 15 November.

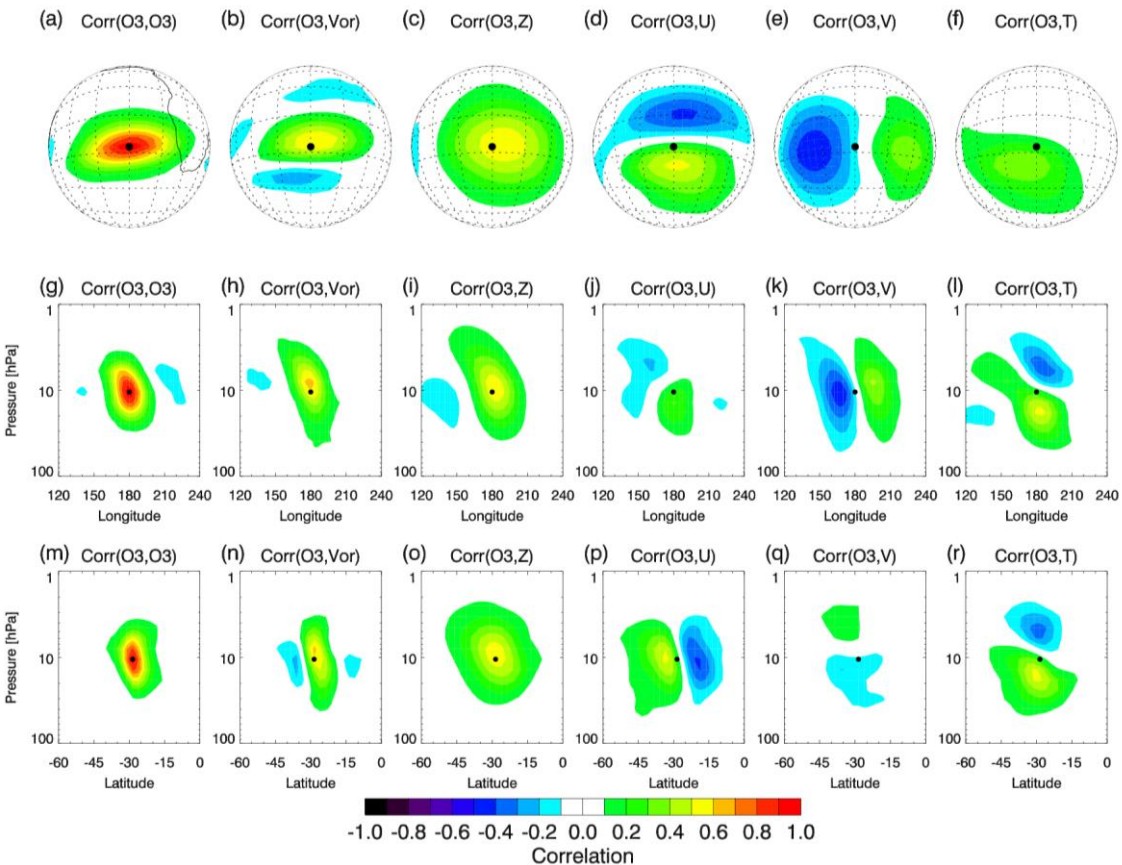

**Fig. 12.** Composite analysis of the ensemble cross-correlations ("Corr") between ozone and other variables on 30 November 2014 (00Z). Calculation is an average at 10.5 hPa and latitude of 28.6°S, and the observation (black dot) is centered at 180° longitude (see text for details). Top row is the horizontal correlation using a satellite projection with grid lines at 10° spacing in longitude and latitude, and continental outline is shown for southern Africa on panel (a). Middle row is the longitude-pressure cross-section, and bottom row is the latitude-pressure cross-section. Columns indicate correlation between ozone and (1) ozone, (2) vorticity, (3) geopotential height, (4) zonal wind, (5) meridional wind, and (6) temperature. Colors are correlation with red (blue) indicating high (low) values.

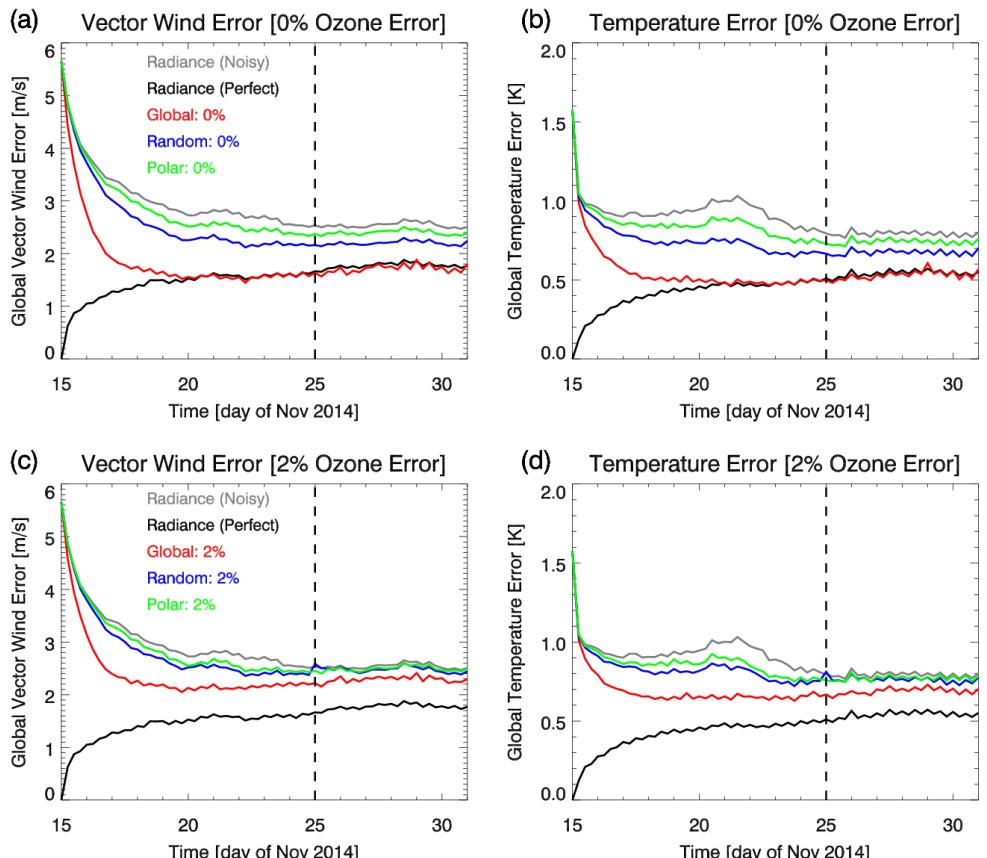

**Fig. 13.** Globally and vertically (77 hPa to 1.1 hPa) averaged RMS errors for five different cycling experiments from 15 November - 1 December 2014, calculated every 6 hours. Black (grey) lines indicate experiments that assimilate perfect (noisy) radiance data only. Colored lines are for experiments that assimilate noisy radiance data along with global (red), random (blue), or polar-orbiting ozone data (green). Panels (a) and (c) show vector wind errors [ms$^{-1}$] for cases with perfect ozone (0% added noise) and noisy ozone (2% added noise), respectively. Panels (b) and (d) show temperature errors [K] for cases with perfect ozone and noisy ozone, respectively. Dashed lines indicate the end of the experimental spin-up phase. See Table 1, Sect. 5.1 and Sect. 5.2.

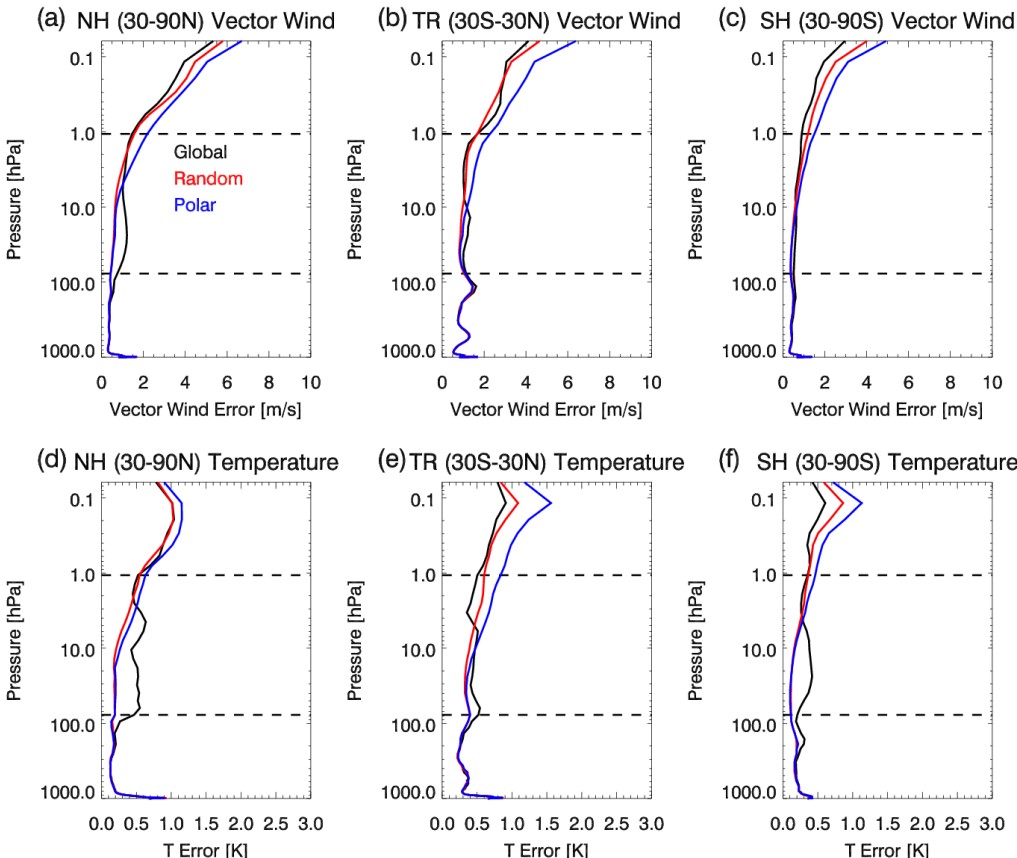

5 **Fig. 14.** RMS vector wind errors [ms$^{-1}$] for (a) NH, (b) TR, and (c) SH and RMS temperature errors [K] for the (d) NH, (e) TR, and (f) SH averaged from 25 November - 1 December 2014 for experiment that assimilated perfect global (black), random (red), and polar-orbiting (blue) ozone data with unperturbed initial conditions. Horizontal dashed lines indicate vertical range of assimilated stratospheric ozone observations. See Table 1, Sect. 4.1.

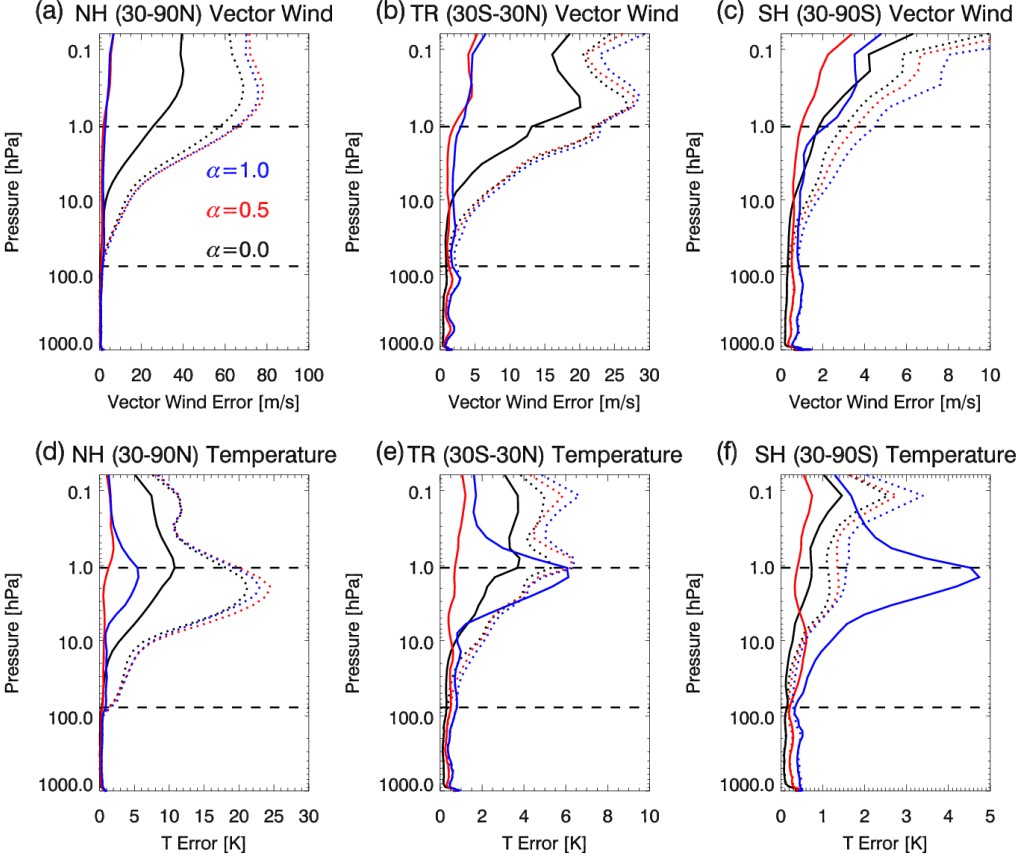

**Fig. 15.** RMS vector wind errors [ms$^{-1}$] for (a) NH, (b) TR, and (c) SH and RMS temperature errors [K] for the (d) NH, (e) TR, and (f) SH averaged from 25 November - 1 December 2014. Solid (dotted) lines are for ozone assimilation experiments (baselines) with α = 0.0 (black), 0.5 (red), and 1.0 (blue). Horizontal dashed lines indicate vertical range of assimilated stratospheric ozone observations. Note that the ranges of the horizontal axes for each panel varies based on maximum baseline errors. See Table 1, Sect. 4.2.

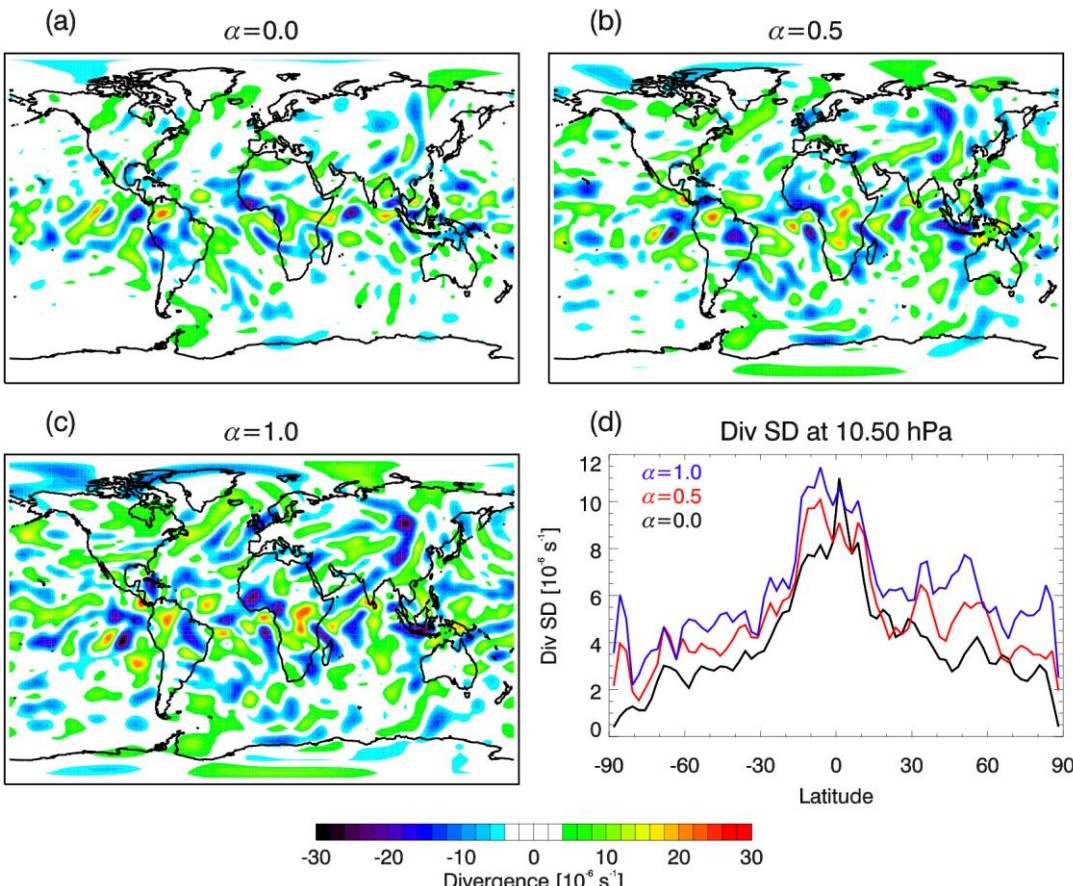

**Fig. 16.** Horizontal maps at 10.5 hPa of the divergence [s⁻¹] on 1 December 2014 (00Z) for ozone assimilation experiments with (a) α = 0.0, (b) α = 0.5, and (c) α = 1.0. Colors show divergence with red (blue) indicating high (low) values. (d) The standard deviation (SD) of the divergence as a function of latitude at 10.5 hPa for 1 December 2014 (00Z). Lines are α = 0.0 (black), α = 0.5 (red), and α = 1.0 (blue). See Table 1, Sect. 4.2.

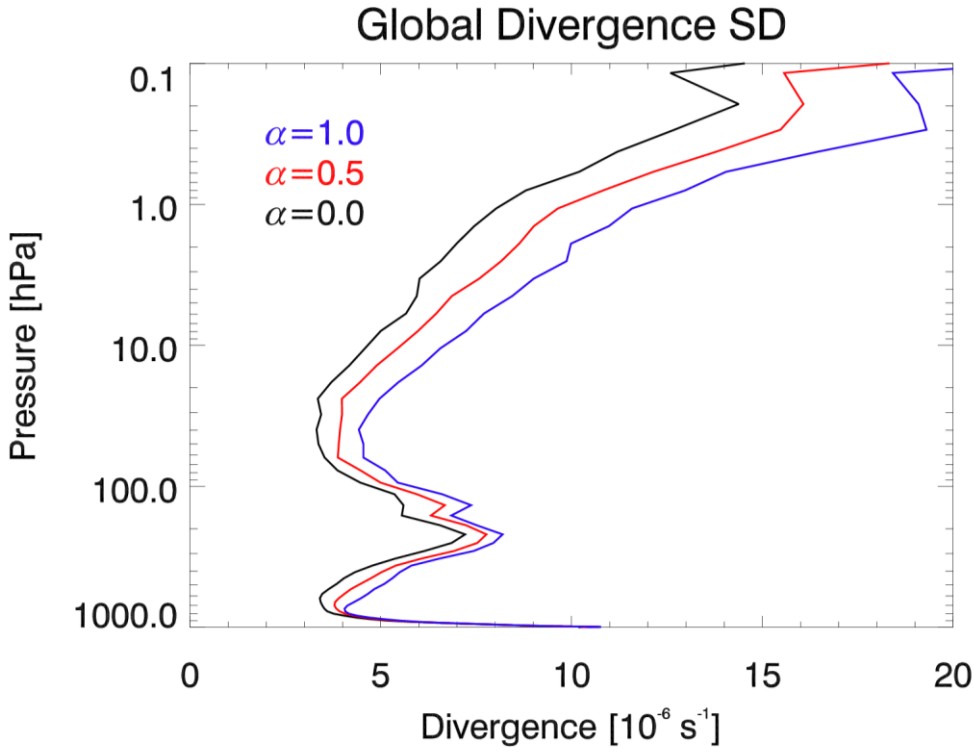

**Fig. 17.** Vertical profiles of the global standard deviation of the divergence [s$^{-1}$] for ozone assimilation experiments on 1 December 2014 (00Z). Lines are α = 0.0 (black), α = 0.5 (red), and α = 1.0 (blue). See Table 1, Sect. 4.2.

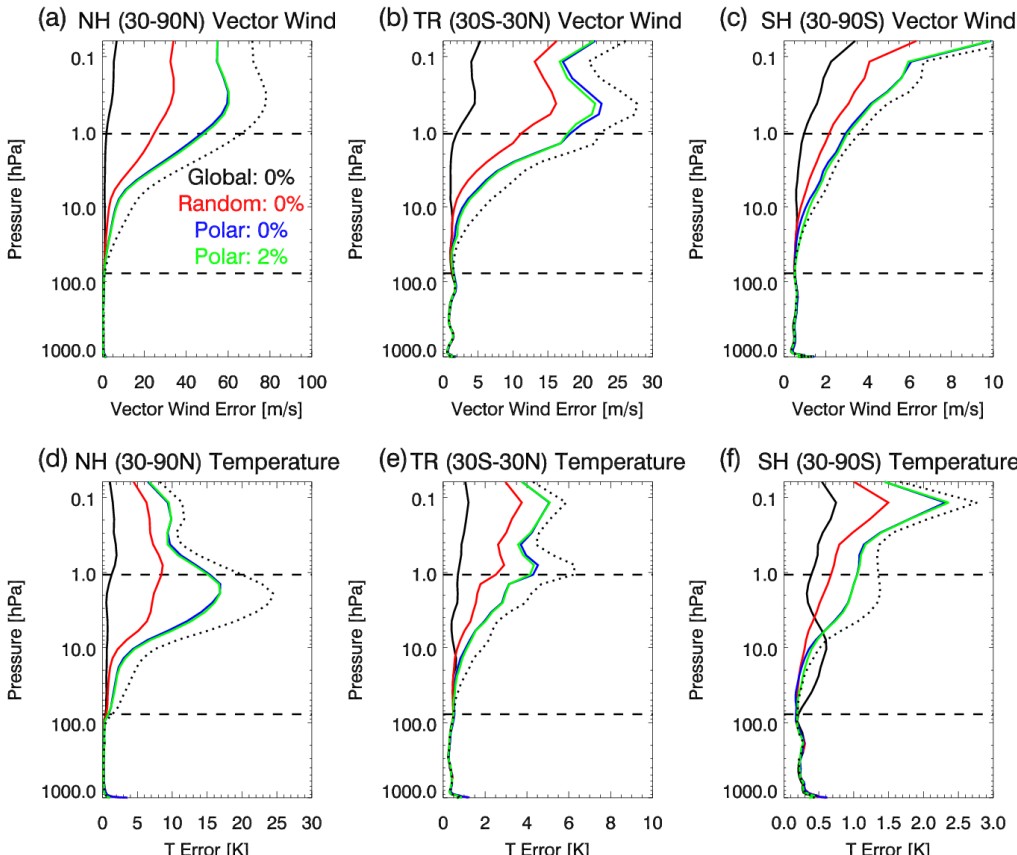

**Fig. 18.** RMS vector wind errors [ms$^{-1}$] averaged from 25 November - 1 December 2014 for (a) NH, (b) TR, and (c) SH and RMS temperature errors [K] for the (d) NH, (e) TR, and (f) SH. Solid lines are for ozone assimilation experiments for global: 0% error (black), random: 0% error (red), polar: 0% error (blue), and polar: 2% error (green). Dotted line is the background for the ozone assimilation experiments. Horizontal dashed lines indicate vertical range of assimilated stratospheric ozone observations. Note that the ranges of the horizontal axes for each panel varies based on maximum baseline errors. See Table 1, Sect. 4.3 and 4.4.

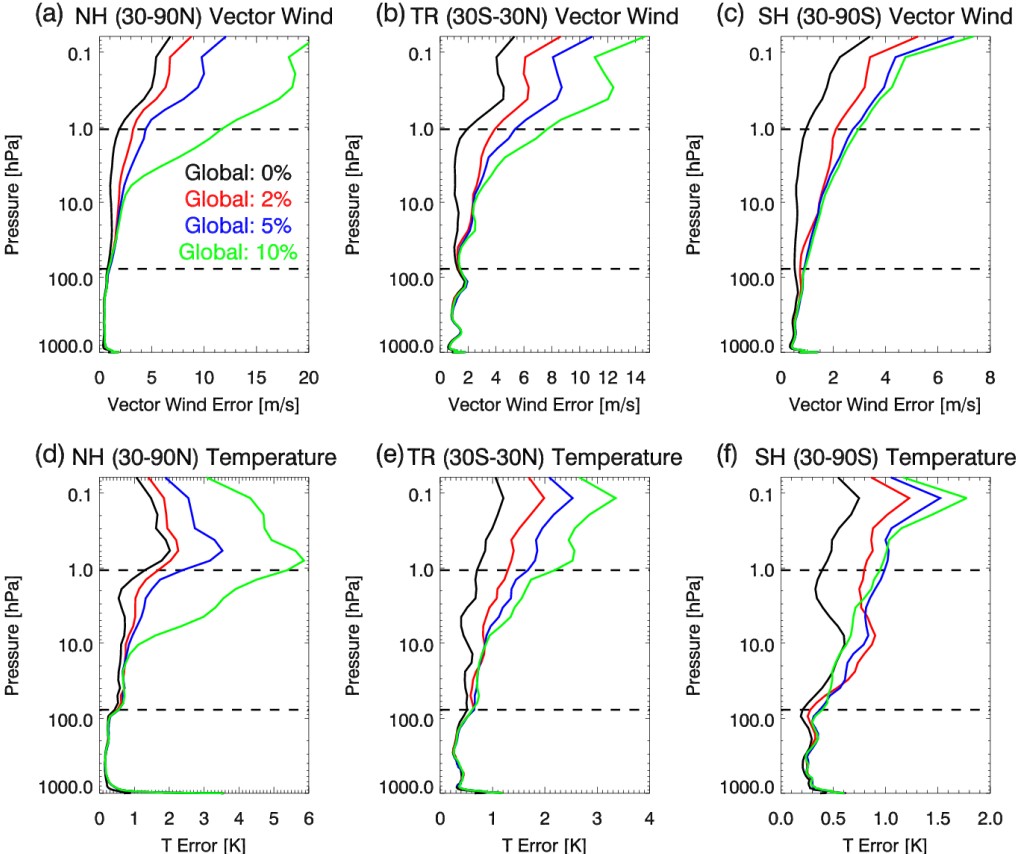

**Fig. 19.** RMS vector wind errors [ms$^{-1}$]  averaged from 25 November - 1 December 2014 for (a) NH, (b) TR, and (c) SH and RMS temperature errors [K] for the (d) NH, (e) TR, and (f) SH. Lines are for ozone assimilation experiments for global ozone with observation errors of 0% (black), 2% (red), 5% (blue), and 10% (green). Horizontal dashed lines indicate vertical range of assimilated stratospheric ozone observations. Note that the ranges of the horizontal axes for each panel varies based on maximum errors. See Table 1, Sect. 4.4.

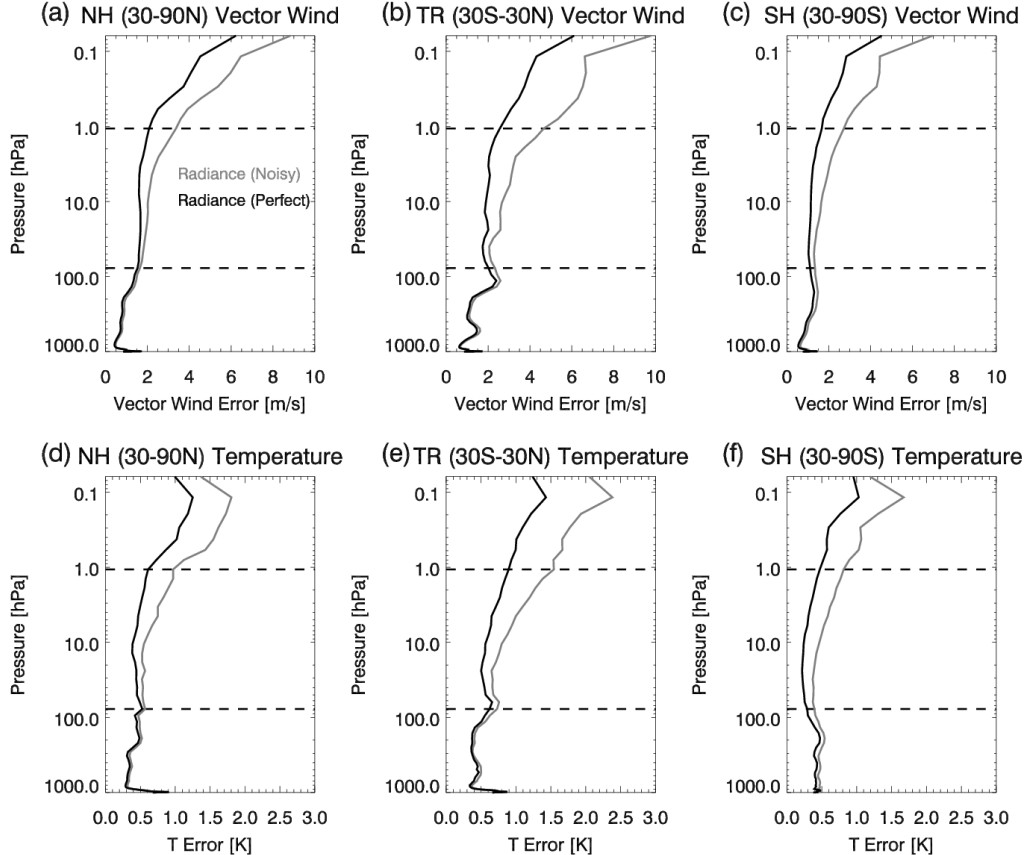

**Fig. 20.** RMS vector wind errors [ms⁻¹] averaged from 25 November - 1 December 2014 for (a) NH, (b) TR, and (c) SH and
RMS temperature errors [K] for the (d) NH, (e) TR, and (f) SH. Black (grey) lines are for assimilation of perfect (noisy)
5 radiance data. Horizontal dashed lines indicate vertical range of assimilated stratospheric ozone observations. Note that the
ranges of the horizontal axes for each panel varies based on maximum errors. See Table 1, Sect. 5.1.

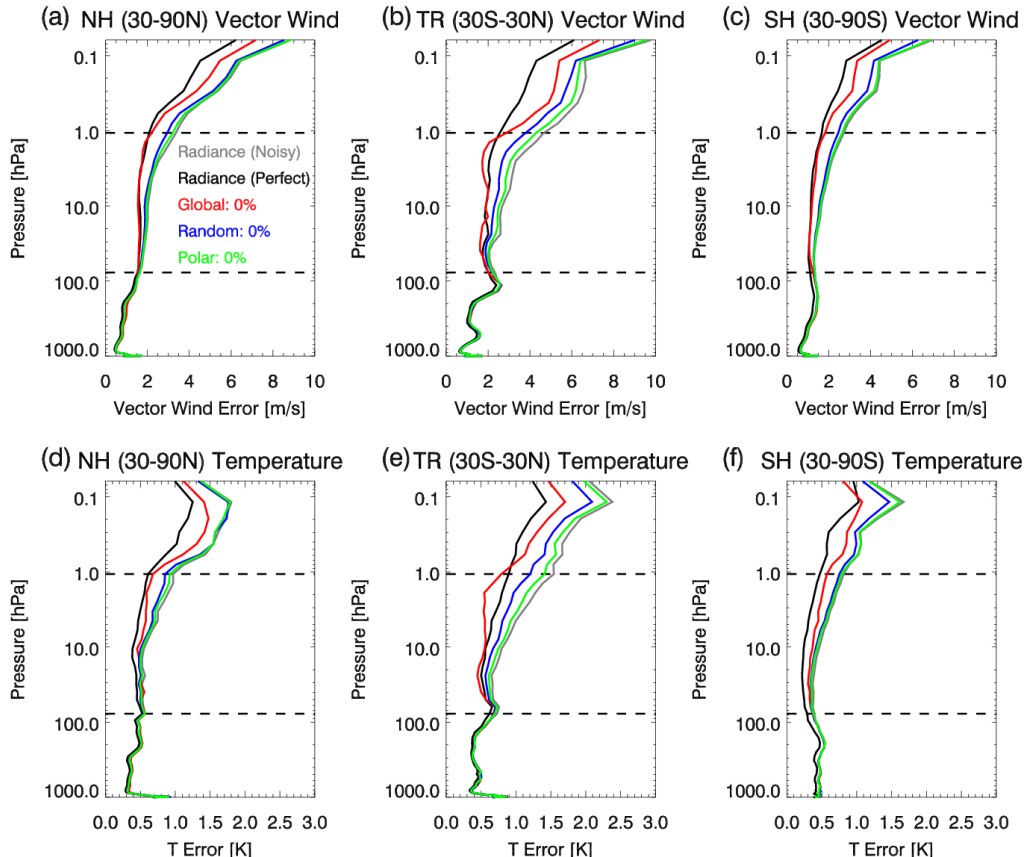

**Fig. 21.** RMS vector wind errors [ms$^{-1}$] averaged from 25 November - 1 December 2014 for (a) NH, (b) TR, and (c) SH and RMS temperature errors [K] for the (d) NH, (e) TR, and (f) SH. Colored lines are for ozone assimilation experiments with noisy radiance and perfect ozone data using sampling pattern of global (red), random (blue), and polar (green). Black (grey) lines are for radiance only assimilation for perfect (noisy) data (same as in Fig. 20). Horizontal dashed lines indicate vertical range of assimilated stratospheric ozone observations. Note that the ranges of the horizontal axes for each panel varies based on maximum errors. See Table 1, Sect. 5.1 and 5.2.

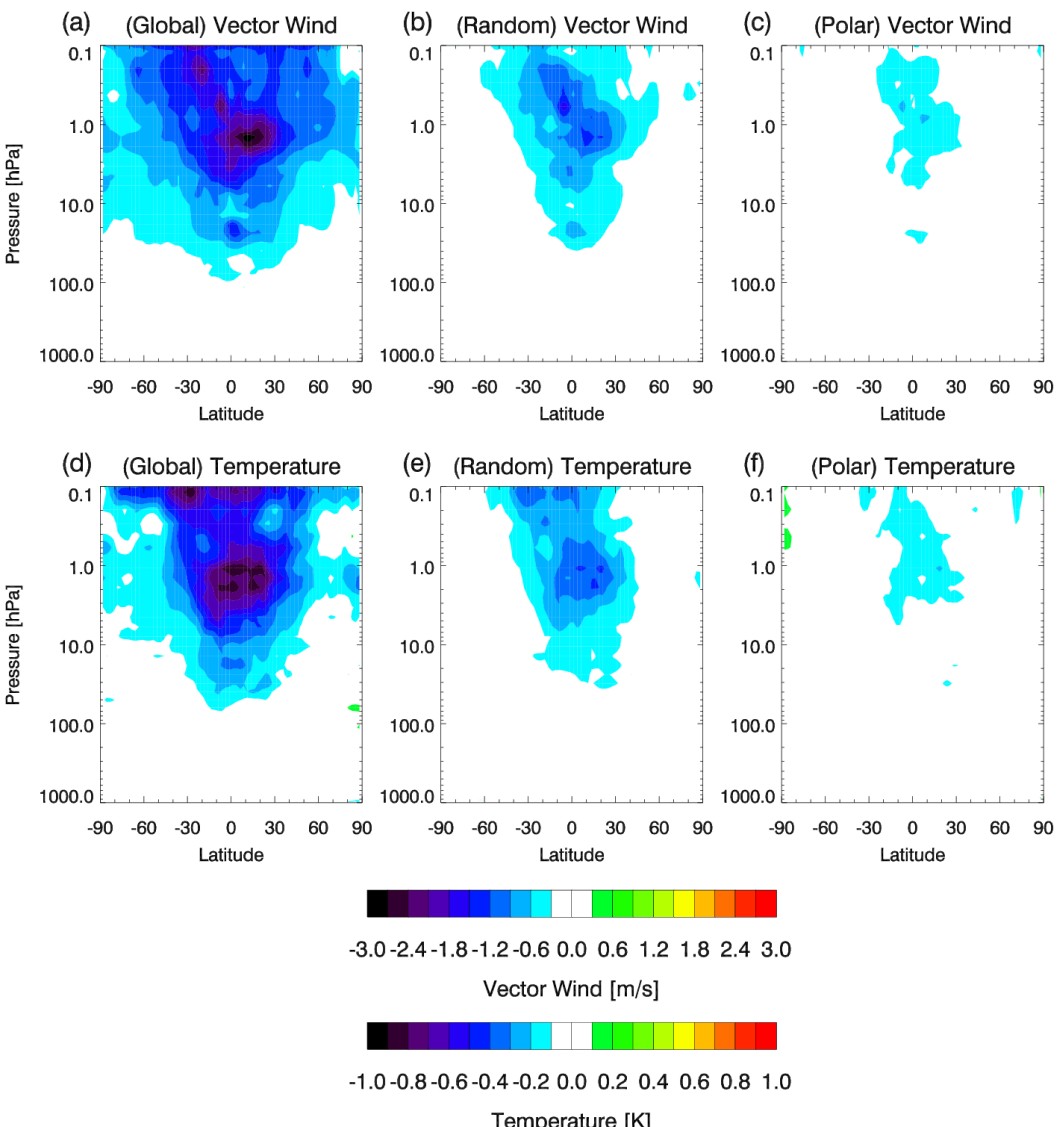

**Fig. 22.** Zonal mean (pressure vs. latitude) plots of the difference in RMS error between assimilation experiments with ozone and radiance data versus those with radiance alone. The results are averaged from 25 November - 1 December 2014. Wind vector results [ms$^{-1}$] are given for perfect ozone (0% imposed error) for (a) global, (b) random, and (c) polar sampling. Temperature results [K] are given for perfect ozone (0% imposed error) for (d) global, (e) random, and (f) polar sampling. See Table 1, Sect. 5.2.

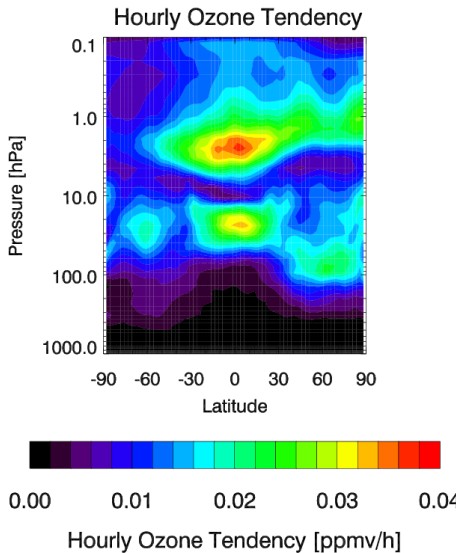

**Fig. 23.** Zonal mean (pressure vs. latitude) plots of the ozone tendency [ppmv/h] for the truth experiment, averaged over forecasts from 0-1 h, 1-2 h, 2-3 h, 3-4 h, 4-5 h, and 5-6 h for 25 November - 1 December 2014. Colors show tendency with red (blue) indicating high (low) values.

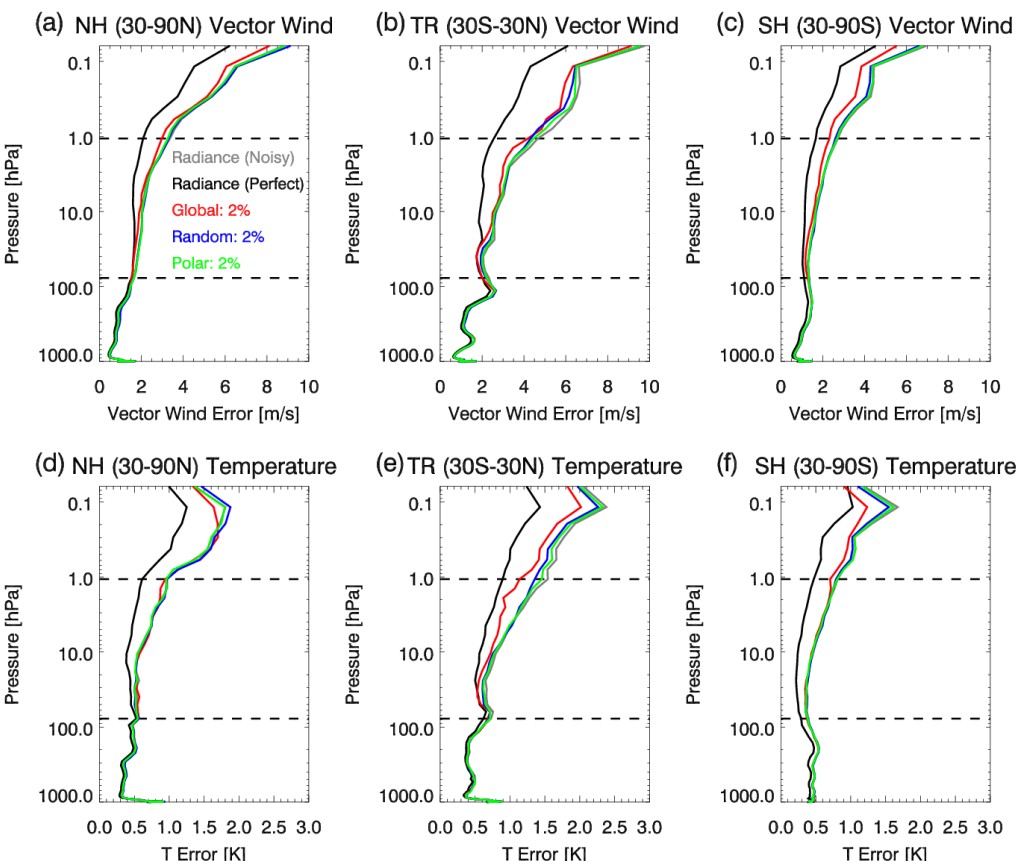

**Fig. 24.** RMS vector wind errors [ms$^{-1}$] averaged from 25 November - 1 December 2014 for (a) NH, (b) TR, and (c) SH and RMS temperature errors [K] for the (d) NH, (e) TR, and (f) SH. Colored lines are for ozone assimilation experiments with noisy radiance and ozone data with 2% error using sampling pattern of global (red), random (blue), and polar (green). Grey lines are for radiance only assimilation for noisy and perfect data (same as in Fig. 20). Horizontal dashed lines indicate vertical range of assimilated stratospheric ozone observations. Note that the ranges of the horizontal axes for each panel varies based on maximum errors. See Table 1, Sect. 5.1 and 5.2.

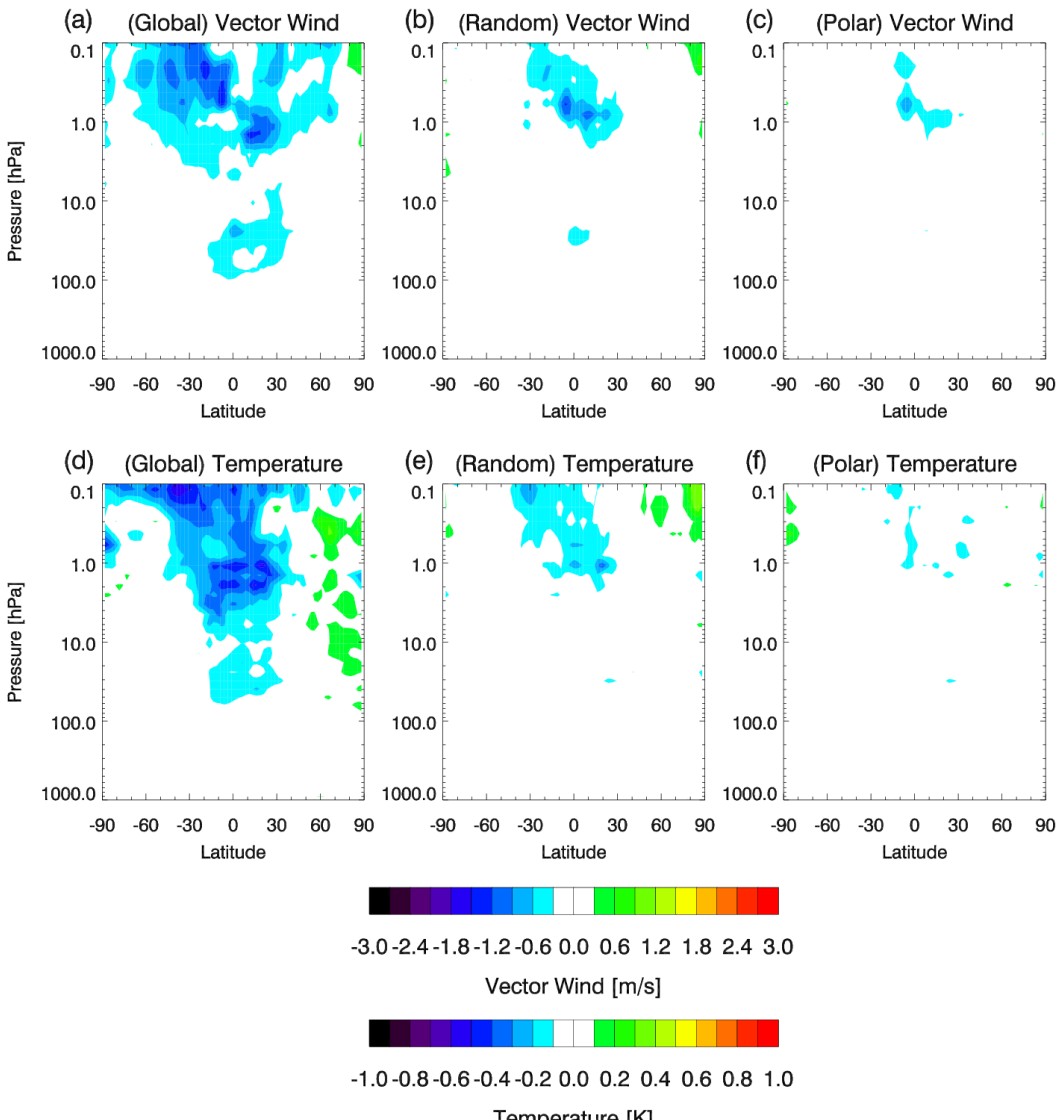

**Fig. 25.** Zonal mean (pressure vs. latitude) plots of the difference in RMS error between assimilation experiments with ozone and radiance data versus those with radiance alone. The results are averaged from 25 November - 1 December 2014. Wind vector results [ms$^{-1}$] are given for noisy ozone (2% imposed error) for (a) global, (b) random, and (c) polar sampling. Temperature results [K] are given for noisy ozone (2% imposed error) for (d) global, (e) random, and (f) polar sampling. See Table 1, Sect. 5.2.