# Peer review of "${\bf Extraction\ of\ wind\ and\ temperature\ information\ from\ hybrid\ 4D-Var} \\ assimilation\ of\ stratospheric\ ozone\ using\ NAVGEM$"

_Atmospheric Chemistry and Physics, 2017_

## Referee Comment (RC1) · A.J. Geer (Referee) · 1 Nov 2017

This is an interesting development in a series of studies by the authors investigating the possibility of wind extraction from tracers in the stratosphere. The latest study moves from using a shallow water model to a full operational-quality NWP system, albeit in an OSSE configuration with a restricted set of observations. The study is well presented and interesting, and the OSSE framework seems novel.

Major comments

1) The introduction needs to be improved in order to summarise the reasons why ozone

assimilation has not yet been successful in operational NWP systems. A short recap of the points made in Allen et al. (2015) would be useful here, and it is worth restating that one of the big problems seems to have been bias between model and observations, rather than particularly the deficiencies in the data assimilation framework. The introduction also needs to motivate the current study better, for example justifying why, since real MLS ozone observations are available, the framework of an OSSE been chosen. It would be useful to state here what benefit NAVGEM gets from MLS ozone assimilation, and to recap any studies that might have been done. A number of minor issues relate to these issues in the introduction:

- Page 2 Line 27/29: "Theoretical studies" may not be the best description of work like that of Peubey and McNally (2009) which tested a real operational NWP system

- Page 2 Line 32: "the operational benefit has not yet been obtained" needs to precisely relate to stratospheric tracers, rather than all tracers, as NWP centres routinely benefit from 4D-Var tracing of tropospheric water vapour from IR and MW radiances, as explained by Peubey and McNally (2009)

2) Based on shallow-water results, line 10 of the introduction says roughly that "including cross-correlations in B between ozone and other variables provides additional ozone-wind benefit". I would have expected to see experiments in this new study that would have explored whether this remains true in the full NAVGEM framework.

3) The OSSE design is described and tested in secs. 2.4 and 4.1. The approach of using real observations in the troposphere seems novel, so it deserves more (critical) investigation within this study. If there are any precedents to this design, they should be cited. There is one interesting parallel with the work of Harnisch et al. (2013) which also used a mixture of real and simulated observations, albeit in an EDA framework.

The statements in sec. 2.4 that the troposphere is "constant, ... essentially independent" and "this gives ... nearly identical tropospheric analyses for each experiment" are imprecise and omit a key idea for understanding this framework. That idea is later supplied in sec. 4.1: "additional observations ... change the numerics of the cost function ... resulting in slight changes to the troposphere". I have run some experiments (Geer, 2016) that explore exactly this issue: when assimilating an identical set of observations into an identical data assimilation system, even the slightest numerical perturbation will generate chaotic divergence between different runs of the data assimilation system. As with my experiments, the troposphere in these new OSSE experiments is not fully constrained by observations and hence will exhibit substantial chaotic variation from one run to the next, whenever the slightest numerical difference is introduced. This spread is somewhat smaller than the expected analysis error for reasons explained in Geer (2016) but is still appreciably large. The troposphere in each separate experiment can be seen as being drawn from a potential ensemble of tropospheric analysis states. The really key understanding is that the stratosphere in the truth experiment (TE) must also be just one realisation drawn from a potential ensemble of stratospheric states, this ensemble being constrained by the data being assimilated in the troposphere. Here also the parallel to Harnisch et al. (2013) is clear.

Thinking this way allows some of the more intriguing results of this study to be analysed better. For example the tropospheric errors shown in Fig. 9 will likely be a rough estimate of the spread of this "tropospheric ensemble" - this would help the authors to explain what they are already saying in Sec. 4.1 about the minimum possible errors in the stratosphere. However it is thus intriguing that by assimilating stratospheric radiance observations simulated from the TE as in Sec. 5.1, Fig. 15, this tropospheric "spread" gets larger, from around 0.2K to 0.5K. This suggests non-optimalities in the NAVGEM data assimilation system or problems in the OSSE framework that need further investigation. Assimilating observations in an optimal system should make errors go down, not up. For me a likely explanation is that the troposphere and the stratosphere are not in reality independent. In the stratospheric radiance assimilation experiment, the troposphere is not identical to that in the truth experiment, and the stratospheric state most compatible with that tropospheric realisation is different to the stratospheric state in the TE. Hence, by trying to make the experiment stratosphere fit that in the TE, it may

require the generation of incorrect increments in the troposphere, which then increase the "spread". Sub-optimalities in the NAVGEM system are also a possibility.

4) In section 4.1, line 10, the expectation of "zero analysis error relative to the TE" with "perfect stratospheric observations" is not so obvious, for a number of reasons:

a) The stratospheric analyses are, even in the presence of near-perfect observations, still just realisations from a hypothetical ensemble of possible stratospheric analyses - this is equivalent to what the authors already say in section 4.1

b) The 0.1 ppmv observation error is not zero, and hence does permit some additional spread within this hypothetical ensemble

c) there may be sub-optimalities of the data assimilation system, such as sampling error in the ensemble-derived part of the background error covariances

d) It is not clear that the stratospheric state is fully determined by the ozone field.

I agree with the authors that their experiments in section 4.1 define a minimum "limit on the level of errors we can reliably distinguish from the TE" but the reasons are more complex than currently stated. It would also be good to see the maximum possible level of error in this experiment, i.e. the "errors" (better, spread) between two realisations of the truth experiment, generated for example by starting another TE from perturbed initial conditions. These minimum and maximum errors roughly define the limits of sensitivity of this novel OSSE design.

5) An additional limitation to the sensitivity of this experimental design is the statistical significance of differences between experiments. In an ideal world all the relevant figures should have statistical significance bars added. For example, intriguing results like the decrease in analysis errors when ozone observation errors are increased (page 14, lines 11-13) could possibly be explained by a lack of statistical significance. However, the statistical significance would not be easy to estimate in this framework except by using an ensemble of perturbed experiments similar to what I was using in Geer

(2016), and analogous to the "spread" between experiments hinted at by Figure 9.

Considerations of sensitivity and statistical significance are important to the conclusion in sec. 5.2 that "in the presence of realistic radiance observations, it is likely that adding ozone assimilation from current ozone retrieval observations .. will have little impact". Again going back to Geer (2016), it is hard to detect the impact of small changes in the observing system in the forecast quality of an operational-quality NWP system. As shown there, adding a single new instrument in the troposphere will only become statistically significant in an experiment containing around 300 forecast samples. In the work under review, there is only a single sample, so it cannot hope to have the required sensitivity or statistical significance. This means the conclusion is unnecessarily pessimistic. If the OSSE were continued for several months, and statistics computed from that whole period, the benefit of ozone assimilation might be seen to be statistically significant. (Many other results presented in this study are more convincing and would likely be statistically significant, but this one is probably not.)

6) Page 16, line 11-14: For similar reasons to those explained in comment 5, the conclusion that current NWP systems can't benefit from ozone-wind interaction in a 4D-var system is probably incorrect. It is more a matter of quantifying the size of that benefit, which is something this current OSSE presentation does not have the sensitivity to explore.

Minor comments

Page 3, Lines 12-13, from "will perturb..." are difficult to follow and need rewriting for clarity.

Page 4, line 27: it is confusing to refer to X' first as an "ensemble state" and then as a "perturbation from the ensemble mean".

Page 5, line 8: "the standard suite" of observations would nowadays include satellite radiances, so this would be better described as a "baseline" or "conventional only" suite

of observations.

Page 6, line 26: Figure 3 caption is missing this piece of information: the level is 10.5hPa

Page 6, line 31-32: That the ozone initial conditions are biased with respect the ozone scheme in the experiments seems a major flaw and needs more explanation or investigation - probably it is no real problem, as implied by section 4.1, that the initial conditions don't matter to the results on 1 December.

Page 7, line 10: "By using a different stratospheric analyses" would be clearer if it was written "by replacing the initial conditions with a different stratospheric analysis"

Page 9, line 8: The high ensemble spread in the vortex is non-intuitive and deserves some explanation. How is this being generated? It should be fairly clear if, for example, the central location of the vortex varies quite a lot across the ensemble.

Page 10, line 10: "It is likely that radiances are the limiting factor..." - please explain this better - I don't understand it.

Citations

Allen, D. R., Hoppel, K. W., and Kuhl, D. D.: Wind extraction potential from ensemble Kalman filter assimilation of stratospheric ozone using a global shallow water model, Atmos. Chem. Phys., 15, 5835-5850, https://doi.org/10.5194/acp-15-5835-2015, 2015.

Geer, Alan J. "Significance of changes in medium-range forecast scores." Tellus A: Dynamic Meteorology and Oceanography 68, no. 1 (2016): 30229, http://dx.doi.org/10.3402/tellusa.v68.30229

Harnisch, F., S.B. Healy, P. Bauer, and S.J. English, 2013: Scaling of GNSS Radio Occultation Impact with Observation Number Using an Ensemble of Data Assimilations. Mon. Wea. Rev., 141, 4395–4413, https://doi.org/10.1175/MWR-D-13-00098.1

Peubey, C., and A. P. McNally. "Characterization of the impact of geostationary clear‐sky radiances on wind analyses in a 4D‐Var context." Quarterly Journal of the Royal Meteorological Society 135, no. 644 (2009): 1863-1876.

---

## Referee Comment (RC2) · T. Milewski (Referee) · 2 Nov 2017

This article addresses the potential of improving wind and temperature analyses in the stratosphere and mesosphere through the assimilation of ozone observations in a reduced-resolution NWP model. It follows up on previous studies that investigated this potential for a variety of data assimilation systems in a simpler model (e.g. global shallow water model), which pointed towards the quality of the Hybrid (covariances) 4D-VAR for this particular purpose. This study is a significant step forward in that it continues to investigate this outstanding question in a more realistic, closer to operational NWP DA setting.

[Figure]

Specifically, this study is an OSSE that focuses on the assimilation of stratospheric ozone observations and its potential added-value over more traditional radiance assimilation. The overall qualities of the study are the well-prepared experimental setup, with a clear progression between experiments, the tests in sensitivity to different parameters and the insights given about the impact of ozone assimilation on the other analyzed variables. However, in the reviewer's opinion, some aspects need to be improved for the article to be ready for final publishing.

Major comments:

The authors are making negative conclusions on the potential benefit of ozone assimilation from the diagnostics of a single case (Dec 1, the final date of a 14-day experiment). There is generally high quality in the experimental setup and the angle of analysis in this study, but it is difficult to objectively distinguish between the random noise in the results and an actual robust signal, in order to draw general conclusions. If the authors intend this article to be a case study, it needs to be firmly stated in the abstract/introduction/conclusions, a more detailed analysis of the current conditions and error patterns, and more caution in making conclusions are needed. Otherwise, the authors need to be more convincing on how this case is representative of more general conditions or, even better, extend the length of the experiments and provide time-averaged results, with statistical significance tests.

Minor comments:

Section 2.1, line 31: "low resolution of T47", please compare it to the operational resolution. This is important considering that this study is addressing the potential benefit of assimilating ozone in NWP systems. Also, How does the reduced resolution of the model might affect the results of radiance assimilation versus ozone assimilation ? In other words, could a higher resolution in the ensembles and/or the background fields help favor assimilation of ozone profiles versus assimilation of radiances ?

Section 2.2, line 18, "60 vertical levels": maybe specify the number of vertical levels

and approximate vertical resolution in the stratosphere and mesosphere.

Section 2.2, line 25, "with a prescribed estimate of the analysis error variance": how is it estimated in this context ?

Section 2.5, line 10, "the perturbation is performed ... different stratospheric analyses": but presumably valid at the same date ? Please specify.

Section 3.1: Did you look at the temporal evolution of sigma_ens, to make sure that the ensemble system has finished its spinup phase ?

Section 3.2: What motivated this choice of latitude and height ? You state that the PV charge analogy is particularly valid in regions "where strong ozone gradients and geostrophic balance occurs", but 28.6S is not a typical region for these two criteria.

Section 4.1: In comparing the ozone-assimilation experiments with perturbed and un-perturbed initial conditions, you are also perturbing the troposphere, which can roughly be considered as a lower boundary condition in your experimental setup. The title of the section "dependence on initial conditions" might be a bit limited or ambiguous. The baseline experiment RMS errors are more representative of the dependence on initial conditions only.

P17, line 9: please correct "The mechanisms through which".

---

## Author Comment (AC1) · 10 Jan 2018

Please see pdf supplement for final response

Please also note the supplement to this comment:
https://www.atmos-chem-phys-discuss.net/acp-2017-802/acp-2017-802-AC1-supplement.pdf

---

## Author Response (AR1)

**Reviewer comments in black, our responses in red. A list of changes made to the figures is included after the responses. The manuscript showing tracked changes is appended at the end.**

**Response to RC1 from Alan Geer, 1 Nov 2017**

This is an interesting development in a series of studies by the authors investigating the possibility of wind extraction from tracers in the stratosphere. The latest study moves from using a shallow water model to a full operational-quality NWP system, albeit in an OSSE configuration with a restricted set of observations. The study is well presented and interesting, and the OSSE framework seems novel.

**Major comments**

1) The introduction needs to be improved in order to summarise the reasons why ozone assimilation has not yet been successful in operational NWP systems. A short recap of the points made in Allen et al. (2015) would be useful here, and it is worth restating that one of the big problems seems to have been bias between model and observations, rather than particularly the deficiencies in the data assimilation framework. The introduction also needs to motivate the current study better, for example justifying why, since real MLS ozone observations are available, the framework of an OSSE been chosen. It would be useful to state here what benefit NAVGEM gets from MLS ozone assimilation, and to recap any studies that might have been done.

The introduction has been rewritten to address the reviewer's comments. The modifications include: (1) a brief recap of points made in Allen et al. (2015), (2) statement about impact of bias between model and observations, (3) justification of using simulated rather than real ozone observations, (4) reference to study by Eckermann et al. (2009) on ozone assimilation within NAVGEM, (5) motivation for the use of both real and simulated data, including references to Harnisch et al. (2013) and Tan et al. (2007).

A number of minor issues relate to these issues in the introduction:

- Page 2 Line 27/29: "Theoretical studies" may not be the best description of work like that of Peubey and McNally (2009) which tested a real operational NWP system

Changed "Theoretical studies" to "Various studies" to be more inclusive of all of the referenced studies.

- Page 2 Line 32: "the operational benefit has not yet been obtained" needs to precisely relate to stratospheric tracers, rather than all tracers, as NWP centres routinely benefit from 4D-Var tracing of tropospheric water vapour from IR and MW radiances, as explained by Peubey and McNally (2009)

Changed the discussion to separate the tropospheric studies from the stratospheric studies.

2) Based on shallow-water results, line 10 of the introduction says roughly that "including cross-correlations in B between ozone and other variables provides additional ozone-wind benefit". I would

have expected to see experiments in this new study that would have explored whether this remains true in the full NAVGEM framework.

The experiments in Sect. 4.2 with different alpha coefficients was designed to address this issue. The alpha=0.0 result is for standard 4D-Var (neglecting initial cross-correlations), while non-zero alpha includes the initial cross-correlations. We showed that using non-zero alpha improved the winds and temperatures at the upper levels. We note that the sensitivity to alpha was only performed for the case of global ozone assimilation with no imposed errors. The subsequent experiments all used alpha=0.5. We have not yet tested the sensitivity to alpha for the case of realistic ozone with/without radiances.

3) The OSSE design is described and tested in secs. 2.4 and 4.1. The approach of using real observations in the troposphere seems novel, so it deserves more (critical) investigation within this study. If there are any precedents to this design, they should be cited. There is one interesting parallel with the work of Harnisch et al. (2013) which also used a mixture of real and simulated observations, albeit in an EDA framework.

We thank the reviewer for reference to Harnisch et al. (2013). We include in the revised introduction a discussion of the EDA approach and how it relates to our study. Basically, we attempt to reduce error correlations between simulated and real data by separating the data into two regions: troposphere and stratosphere. The true stratosphere is created by a cycling analysis using only tropospheric data, while the simulated data are only in the stratosphere. As discussed in the introduction, this separation is not perfect, since weighting functions and vertical error correlations can extend across the 100 hPa level. In Section 2.3 we now include plots of analysis increments from the truth experiment to examine the separation in more detail.

The statements in sec. 2.4 that the troposphere is "constant, ... essentially independent" and "this gives ... nearly identical tropospheric analyses for each experiment" are imprecise and omit a key idea for understanding this framework. That idea is later supplied in sec. 4.1: "additional observations ... change the numerics of the cost function ... resulting in slight changes to the troposphere".

We deleted the first sentence quoted and changed the wording on the second sentence to avoid confusion. Further discussion on this point is provided below.

I have run some experiments (Geer, 2016) that explore exactly this issue: when assimilating an identical set of observations into an identical data assimilation system, even the slightest numerical perturbation will generate chaotic divergence between different runs of the data assimilation system. As with my experiments, the troposphere in these new OSSE experiments is not fully constrained by observations and hence will exhibit substantial chaotic variation from one run to the next, whenever the slightest numerical difference is introduced. This spread is somewhat smaller than the expected analysis error for reasons explained in Geer (2016) but is still appreciably large. The troposphere in each separate experiment can be seen as being drawn from a potential ensemble of tropospheric analysis states. The really key understanding is that the stratosphere in the truth experiment (TE) must also be just one realisation drawn from a potential ensemble of stratospheric states, this ensemble being constrained by the data being assimilated in the troposphere. Here also the parallel to Harnisch et al. (2013) is clear.

This is very helpful for analyzing our results. We include a discussion of the tropospheric sensitivity to slight perturbations in Sect. 2.4 and enhance the discussion of Sect. 4.1. In Fig. 14 of that section, we now include three perturbation experiments in which we assimilate perfect global, random, and polar-orbiting data with unperturbed initial conditions.

Thinking this way allows some of the more intriguing results of this study to be analysed better. For example the tropospheric errors shown in Fig. 9 will likely be a rough estimate of the spread of this "tropospheric ensemble" - this would help the authors to explain what they are already saying in Sec. 4.1 about the minimum possible errors in the stratosphere.

We now mention that the errors in this figure (now Fig. 14) are a rough estimate of the spread of a hypothetical "tropospheric ensemble" as well as minimum possible errors in the stratosphere.

However it is thus intriguing that by assimilating stratospheric radiance observations simulated from the TE as in Sec. 5.1, Fig. 15, this tropospheric "spread" gets larger, from around 0.2K to 0.5K. This suggests non-optimalities in the NAVGEM data assimilation system or problems in the OSSE framework that need further investigation. Assimilating observations in an optimal system should make errors go down, not up. For me a likely explanation is that the troposphere and the stratosphere are not in reality independent. In the stratospheric radiance assimilation experiment, the troposphere is not identical to that in the truth experiment, and the stratospheric state most compatible with that tropospheric realisation is different to the stratospheric state in the TE. Hence, by trying to make the experiment stratosphere fit that in the TE, it may require the generation of incorrect increments in the troposphere, which then increase the "spread". Sub-optimalities in the NAVGEM system are also a possibility.

We agree that likely the stratosphere and troposphere are not completely independent. In addition to the perturbation that the radiance observations add to the system, some of the radiance observations are influence by the tropospheric observation. This means the observational errors are not independent, even though they are assumed to be in the DAS. This sub-optimality may be the cause of the increased "spread". We add further discussion on this in Sect. 5.1.

4) In section 4.1, line 10, the expectation of "zero analysis error relative to the TE" with "perfect stratospheric observations" is not so obvious, for a number of reasons:

We changed the discussion in this section as follows.

a) The stratospheric analyses are, even in the presence of near-perfect observations, still just realisations from a hypothetical ensemble of possible stratospheric analyses - this is equivalent to what the authors already say in section 4.1

This is addressed in modifications to Section 4.1. We use the phrase "simple four-member null set" rather than "hypothetical ensemble".

b) The 0.1 ppmv observation error is not zero, and hence does permit some additional spread within this hypothetical ensemble

We emphasized this allows for variations in the realized state.

c) there may be sub-optimalities of the data assimilation system, such as sampling error in the ensemble-derived part of the background error covariances

We included the possibility of sub-optimalities.

d) It is not clear that the stratospheric state is fully determined by the ozone field.

We attempt to show by the perfect global ozone case the extent to which the stratospheric state can be determined by ozone alone. We do not say "fully determined" in the revision, but rather specify the errors from this experiment.

I agree with the authors that their experiments in section 4.1 define a minimum "limit on the level of errors we can reliably distinguish from the TE" but the reasons are more complex than currently stated.

Hopefully the additional discussion in this section helped to clarify our results.

It would also be good to see the maximum possible level of error in this experiment, i.e. the "errors" (better, spread) between two realisations of the truth experiment, generated for example by starting another TE from perturbed initial conditions. These minimum and maximum errors roughly define the limits of sensitivity of this novel OSSE design.

The baseline experiments were designed to provide the maximum level of error. For the ozone-only experiments, the baseline experiment assimilated the same data as the truth, but included perturbed initial conditions in the stratosphere. The resulting errors are provided, for example, in the dashed lines on revised Fig. 15. For the ozone and radiance tests, we included bounds of the minimum and maximum expected errors by using the noisy and perfect radiance assimilation experiments.

5) An additional limitation to the sensitivity of this experimental design is the statistical significance of differences between experiments. In an ideal world all the relevant figures should have statistical significance bars added. For example, intriguing results like the decrease in analysis errors when ozone observation errors are increased (page 14, lines 11-13) could possibly be explained by a lack of statistical significance. However, the statistical significance would not be easy to estimate in this framework except by using an ensemble of perturbed experiments similar to what I was using in Geer (2016), and analogous to the "spread" between experiments hinted at by Figure 9.

Considerations of sensitivity and statistical significance are important to the conclusion in sec. 5.2 that "in the presence of realistic radiance observations, it is likely that adding ozone assimilation from current ozone retrieval observations .. will have little impact". Again going back to Geer (2016), it is hard to detect the impact of small changes in the observing system in the forecast quality of an

operational-quality NWP system. As shown there, adding a single new instrument in the troposphere will only become statistically significant in an experiment containing around 300 forecast samples. In the work under review, there is only a single sample, so it cannot hope to have the required sensitivity or statistical significance. This means the conclusion is unnecessarily pessimistic. If the OSSE were continued for several months, and statistics computed from that whole period, the benefit of ozone assimilation might be seen to be statistically significant. (Many other results presented in this study are more convincing and would likely be statistically significant, but this one is probably not.)

We agree that our conclusions were premature, based on only one time at the end of the experiment. We decided to include 6 days of averaging (following a 10-day spin-up) in all of the revised error profiles in order to make the results more robust. A plot of the time series of errors for several experiments is provided in Fig. 13 to show that the errors are relatively level during the last 6 days. But even with these 25 analyses, it is unlikely that we can reach a conclusion on statistical significance. See further discussion on our altered approach in the response to the second review.

6) Page 16, line 11-14: For similar reasons to those explained in comment 5, the conclusion that current NWP systems can't benefit from ozone-wind interaction in a 4D-var system is probably incorrect. It is more a matter of quantifying the size of that benefit, which is something this current OSSE presentation does not have the sensitivity to explore.

Good point. We altered this discussion accordingly.

**Minor comments**

Page 3, Lines 12-13, from "will perturb..." are difficult to follow and need rewriting for clarity.

We modified this sentence.

Page 4, line 27: it is confusing to refer to X' first as an "ensemble state" and then as a "perturbation from the ensemble mean".

We clarify the wording. X is ensemble state, X' is perturbation from the ensemble mean.

Page 5, line 8: "the standard suite" of observations would nowadays include satellite radiances, so this would be better described as a "baseline" or "conventional only" suite of observations.

Changed "standard suite" to "conventional".

Page 6, line 26: Figure 3 caption is missing this piece of information: the level is 10.5hPa

Included the level in the figure caption (now Figure 6).

Page 6, line 31-32: That the ozone initial conditions are biased with respect the ozone scheme in the experiments seems a major flaw and needs more explanation or investigation - probably it is no real

problem, as implied by section 4.1, that the initial conditions don't matter to the results on 1 December.

We do not think this is a major problem, given the relative insensitivity to initial conditions. We decided to remove the comment about initializing with ozone based on another scheme to avoid confusion.

Page 7, line 10: "By using a different stratospheric analyses" would be clearer if it was written "by replacing the initial conditions with a different stratospheric analysis"

Good suggestion. We change the wording as suggested.

Page 9, line 8: The high ensemble spread in the vortex is non-intuitive and deserves some explanation. How is this being generated? It should be fairly clear if, for example, the central location of the vortex varies quite a lot across the ensemble.

While individual maps of ensemble members show some variability in the location, orientation, and shape of the vortex, the ozone shows even larger variability. We think this is due to slight variations in the vortex evolution in each ensemble member that result in differences in ozone advection, which accumulate with time due to the long photochemical lifetime of ozone in the NH winter polar region. This causes the initially small spread to increase with time over the experiment. The discussion was modified accordingly.

Page 10, line 10: "It is likely that radiances are the limiting factor..." - please explain this better - I don't understand it.

We removed this sentence, since it was not helpful in the discussion.

Citations - these citations are now all included in the manuscript

Allen, D. R., Hoppel, K. W., and Kuhl, D. D.: Wind extraction potential from ensemble Kalman filter assimilation of stratospheric ozone using a global shallow water model, Atmos. Chem. Phys., 15, 5835-5850, https://doi.org/10.5194/acp-15-5835-2015, 2015.

Geer, Alan J. "Significance of changes in medium-range forecast scores." Tellus A: Dynamic Meteorology and Oceanography 68, no. 1 (2016): 30229, http://dx.doi.org/10.3402/tellusa.v68.30229

Harnisch, F., S.B. Healy, P. Bauer, and S.J. English, 2013: Scaling of GNSS Radio Occultation Impact with Observation Number Using an Ensemble of Data Assimilations. Mon. Wea. Rev., 141, 4395–4413, https://doi.org/10.1175/MWR-D-13-00098.1

Peubey, C., and A. P. McNally. "Characterization of the impact of geostationary clearâ˘A ˘ Rsky radiances on wind analyses in a 4Dâ˘A ˘RVar context." Quarterly Journal of the Royal Meteorological Society 135, no. 644 (2009): 1863-1876.

**Response to RC2 from Thomas Milewski, 2 Nov 2017**

This article addresses the potential of improving wind and temperature analyses in the stratosphere and mesosphere through the assimilation of ozone observations in a reduced-resolution NWP model. It follows up on previous studies that investigated this potential for a variety of data assimilation systems in a simpler model (e.g. global shallow water model), which pointed towards the quality of the Hybrid (covariances) 4D-VAR for this particular purpose. This study is a significant step forward in that it continues to investigate this outstanding question in a more realistic, closer to operational NWP DA setting.

Specifically, this study is an OSSE that focuses on the assimilation of stratospheric ozone observations and its potential added-value over more traditional radiance assimilation. The overall qualities of the study are the well-prepared experimental setup, with a clear progression between experiments, the tests in sensitivity to different parameters and the insights given about the impact of ozone assimilation on the other analyzed variables. However, in the reviewer's opinion, some aspects need to be improved for the article to be ready for final publishing.

**Major comments:**

The authors are making negative conclusions on the potential benefit of ozone assimilation from the diagnostics of a single case (Dec 1, the final date of a 14-day experiment). There is generally high quality in the experimental setup and the angle of analysis in this study, but it is difficult to objectively distinguish between the random noise in the results and an actual robust signal, in order to draw general conclusions. If the authors intend this article to be a case study, it needs to be firmly stated in the abstract/introduction/conclusions, a more detailed analysis of the current conditions and error patterns, and more caution in making conclusions are needed. Otherwise, the authors need to be more convincing on how this case is representative of more general conditions or, even better, extend the length of the experiments and provide time-averaged results, with statistical significance tests.

We agree that the general conclusions that we drew from a single case was premature, and therefore are editing the text accordingly (see also response to the first review). We also changed the error profiles to be time-averaged results over the last 6 days of the 16-day experiment, allowing for 10 days of spin-up. The justification for spin-up time is discussed further below. The time-averaging decreases noise from what we had in a single time, but the limited extent of the experiment does not permit testing of statistical significance (see also comments and responses from the first reviewer). We therefore intend this to be a case study, and added further results (see next paragraph) and text to indicate this.

We also included additional discussion of the current conditions and error patterns. Maps of ozone and height in the lower stratosphere (new Fig. 5) and upper stratosphere (new Fig. 7) were added. In addition, we included time series of the globally- and vertically-averaged errors (new Fig. 13), zonal mean plots of errors for the ozone/radiance experiments (new Fig. 22 and 25), and further diagnostics including the ozone tendency (Fig. 23) and the mean increment (Fig. 3). These enhance the paper as a case study of a particular event, providing guidance for future work that would examine the statistical significance of ozone assimilation on the winds and temperatures.

**Minor comments:**

Section 2.1, line 31: "low resolution of T47", please compare it to the operational resolution. This is important considering that this study is addressing the potential benefit of assimilating ozone in NWP systems.

Operational resolution is T425L60 (0.28° Gaussian grid spacing at the equator) for the outer-loop and T119L60 (1.0°) for the inner-loop, which is the resolution of the data assimilation. T47 is 2.5° spacing at the Equator. We include this comparison in the revision.

Also, How does the reduced resolution of the model might affect the results of radiance assimilation versus ozone assimilation? In other words, could a higher resolution in the ensembles and/or the background fields help favor assimilation of ozone profiles versus assimilation of radiances?

It is difficult to address this without actually performing the experiments, and we want to avoid too much speculation. However, we added a brief discussion of the potential impact of higher resolution on ozone assimilation in Sect. 6.

Section 2.2, line 18, "60 vertical levels": maybe specify the number of vertical levels and approximate vertical resolution in the stratosphere and mesosphere.

In the stratosphere (lower mesosphere), there are 18 (7) levels. The vertical spacing ranges from ~1.5 km in the lower stratosphere to ~2.5 km at the stratopause to ~5 km at the model top. This is now indicate in the revision in Section 2.1.

Section 2.2, line 25, "with a prescribed estimate of the analysis error variance": how is it estimated in this context ?

Modified text to include the following description of the analysis error variance. The ET scheme transforms the previous 6-h ensemble perturbations into a new set of initial perturbations such that the initial ensemble covariance is consistent with a prescribed climatological 3D-Var based estimate of

the analysis error variance. The climatological variances are averaged from 10 June 2015 to 10 August 2015, and are the same as those used in the operational NAVGEM system.

Section 2.5, line 10, "the perturbation is performed ... different stratospheric analyses": but presumably valid at the same date ? Please specify.

Yes, these are valid at the same time (15 November 2014, 00Z). We indicate this in the revision.

Section 3.1: Did you look at the temporal evolution of sigma_ens, to make sure that the ensemble system has finished its spinup phase ?

We included analysis of the ensemble spin-up in the revision (see revised Fig. 10). The ensemble spin-up took about 10 days for T and Z, and about 8 days for U and V. We updated several of our results to be time averages over the last 6 days of the experiments, following the spin-up period.

Section 3.2: What motivated this choice of latitude and height ? You state that the PV charge analogy is particularly valid in regions "where strong ozone gradients and geostrophic balance occurs", but 28.6S is not a typical region for these two criteria.

We revised this sentence to avoid confusion. The horizontal ozone gradients are actually relatively large at this latitude for this time period. We added zonal mean maps of ozone to the revised Figure 4 in order to illustrate this (see white dotted line on Fig. 4c). The geostrophic balance at ~30S would be limited somewhat by the Coriolis parameter being around sin(30) = 0.5, but the circulation still resembles geostrophy at this latitude.

Section 4.1: In comparing the ozone-assimilation experiments with perturbed and unperturbed initial conditions, you are also perturbing the troposphere, which can roughly be considered as a lower boundary condition in your experimental setup. The title of the section "dependence on initial conditions" might be a bit limited or ambiguous. The baseline experiment RMS errors are more representative of the dependence on initial conditions only.

We changed the title of this subsection to "Sensitivity of the analysis to perturbations in the DAS," and we added more relevant discussion, also answering some questions from the first reviewer.

P17, line 9: please correct "The mechanisms through which".

We corrected this phrase (changed "though" to "through").

**List of Changes to the Figures**

| New | Original | Changes from Original |
|---|---|---|
| Figure 1 | Figure 1 | None |
| Figure 2 | Figure 2 | None |
| Figure 3 | ---------- | New figure: Analysis increments |
| Figure 4 | Figure 5 | Included zonal mean ozone plots |
| Figure 5 | ---------- | New figure: Ozone maps at 77 hPa |
| Figure 6 | Figure 3 | None |
| Figure 7 | ---------- | New figure: Ozone maps at 1.1 hPa |
| Figure 8 | Figure 4 | None |
| Figure 9 | Figure 6 | None |
| Figure 10 | ---------- | New figure: Ensemble standard deviation time series plots |
| Figure 11 | Figure 7 | Included continents on panels (a) and (g) |
| Figure 12 | Figure 8 | None |
| Figure 13 | ---------- | New figure: Time series of globally and vertically averaged errors |
| Figure 14 | Figure 9 | See 1 below, now show perfect global, random, and polar cases |
| Figure 15 | Figure 10 | See 1 below |
| Figure 16 | Figure 11 | Changed $10^6$ to $10^{-6}$ |
| Figure 17 | Figure 12 | Changed $10^6$ to $10^{-6}$ |
| Figure 18 | Figure 13 | See 1, 2 below, changed dotted line from green to black |
| Figure 19 | Figure 14 | See 1, 2 below, removed dotted lines, changed x-axes |
| Figure 20 | Figure 15 | See 1 below, colors changed |
| Figure 21 | Figure 16 | See 1 below |
| Figure 22 | Figure 17 | See 1 below, replaced line plots with zonal mean differences |
| Figure 23 | ---------- | New figure: Hourly ozone tendency |
| Figure 24 | Figure 18 | See 1, 2 below |
| Figure 25 | Figure 19 | See 1, 2 below, replaced line plots with zonal mean differences |

1. Data shown are now averaged from 25 Nov - 1 Dec 2014, rather than only at 1 Dec 2014.

2. For the experiments that assimilated "noisy" ozone data, we decided that in the specification of the observation error standard deviation that we limit the minimum value to 0.1 ppmv, the same number used for the "perfect" ozone (see Table 1), rather than use the same percent value. This was to avoid having the system pull too tightly to very precise data in the lower stratosphere.

[revised manuscript text omitted]